# Spatially resolved soil solution chemistry in a central European atmospherically polluted high-elevation catchment

Daniel A. Petrash[1], Frantisek Buzek[1], Martin Novak[1], Bohuslava Cejkova[1], Pavel Kram[1], Tomas Chuman[1], Jan Curik[1], Frantisek Veselovsky[2], Marketa Stepanova[1], Oldrich Myska[1], Pavla Holeckova[1], Leona Bohdalkova[1]

[1] Czech Geological Survey, Department of Environmental Geochemistry and Biogeochemistry, Geologicka 6, 152 00, Prague 5, Czech Republic

[2] Czech Geological Survey, Department of Rock Geochemistry, Geologicka 6, 152 00, Prague 5, Czech Republic

**Abstract.** We collected soil solutions by suction lysimeters in a central European temperate forest with a history of acidification-related spruce die-back in order to interpret spatial patterns of soil nutrient partitioning, compare them with stream water chemistry and evaluate these parameters relative to concurrent loads of anions and cations in precipitation. Five lysimeter nests were installed in the 33-ha U dvou loucek (UDL) mountain catchment at different topographic positions (hilltops, slopes and valley). Following equilibration, monthly soil solution samples were interrogated over a 2-year period with regard to their $SO_4^{2-}$, $NO_3^-$, $NH_4^+$, $Na^+$, $K^+$, $Ca^{2+}$, $Mg^{2+}$ and total dissolved Al concentrations, organic carbon (DOC) and pH. Soil pits were excavated in the vicinity of each lysimeter nest to also constrain soil chemistry. For an estimation of phosphorus (P) availability, ammonium oxalate extraction of soil samples was performed. Cation exchange capacity (CEC $\leq$ 58 meq kg$^{-1}$) and base saturation (BS $\leq$ 13 %) were found to be significantly lower at UDL than in other monitored central European small catchments areas. Spatial trends and seasonality in soil solution chemistry support belowground inputs from mineral-stabilized legacy pollutants. Overall, the soil solution data suggest that the ecosystem was still chemically out of balance relative to the concurrent loads of anions and cations in precipitation, documenting incomplete recovery from acidification. Nearly 30 years after peak acidification, UDL exhibited similar soil solution concentrations of $SO_4^2$, $Ca^{2+}$ and $Mg^{2+}$ as median values at the Pan-European International Co-operative Program (ICP) Forest sites with similar bedrock lithology and vegetation cover, yet $NO_3^-$ concentrations were an order of magnitude higher. When concentrations of $SO_4^{2-}$, $NO_3^-$ and base cations in runoff are compared to soil pore waters, higher concentration in runoff point to lateral surficial leaching of pollutants and nutrients in excess than from topsoil to subsoil. With P availability being below the lowest range observed in soil plots of the Czech Republic, the managed forest ecosystem in UDL probably reflects growing inputs of C from regenerating vegetation in the N-saturated soil, which leads to P depletion in the soil. In addition, the observed spatial variability provides evidence pointing to substrate variability, C and P bioavailability, and landscape as major controls over base metal leaching toward the subsoil level in N-saturated catchments.

## 1 Introduction

During the second half of the 20th century, atmospheric deposition of reactive nitrogen (N) and sulfate ($SO_4^{2-}$) caused persistent perturbations in temperate forest soils and water sheds across Europe (Blazkova, 1996; Alewell et al., 2001; Verstraeten et al., 2017). Reaction of sulfur dioxide ($SO_2$) and nitrogen oxides ($NO_x$) with water molecules in the atmosphere produced sulfuric and nitric acids ($H_2SO_4$, $HNO_3$) which then entered mountain forest ecosystems *via* wet and dry deposition. Intergovernmental cooperation to significantly abate detrimental emissions was implemented during the 1980s and resulted in a decrease in the atmospheric deposition of soil-acidifying species. Industry restructuring and installation of scrubbers in power plants significantly reduced industrial $SO_2$ emission rates by more than 90% in one of the most polluted industrial regions of central Europe, known as the "Black Triangle", which includes mountainous border regions of three countries: Czech Republic, East Germany and Poland (Fig. 1a; Blazkova, 1996; Novak et al., 2005). A decrease in soil solution $SO_4^{2-}$ concentrations followed, and has progressively lowered soil solution ionic strengths while contributing to a progressive raise in soil pH, which may have in turn increased organic matter leaching by lowering the soil solution concentrations of aluminum ions ($Al^{3+}$) (*e.g.*, Monteith et al., 2007).

Although also significantly decreased, atmospheric inputs of reactive N species in excess to the nutritional demands of plants and microorganisms have prevailed (Waldner et al., 2014, 2015). These have resulted in forest ecosystem perturbations consisting on a cascade of biogeochemical reactions linked to soil N-saturation (Galloway et al., 2003). For instance, despite a ~40 % reduction in atmospheric N inputs (Kopacek et al., 2013), in the Czech Republic alone the value of the total nitrogen deposition remained—during the last decade—in the range of 70,000–80,000 t per year as a result of the production of $NO_x$ emissions from transport, industrial production and energy generation (CENIA, 2017). The unchanged figures despite significant efforts for controlling industrial emissions over the same period is a societal concern since there is growing uncertainty on whether or not central European temperate forest ecosystems will be in capacity to continue acting as major sinks for a 15-20% intensification in anthropogenic emissions of reactive N to the atmosphere (Galloway et al. 2008). A detrimental effect of unmanaged soil N-saturation in catchment areas is the propagation of environmental effects to nearby lacustrine ecosystems (Kopacek et al. 1995).

N-saturation in soils is indicated by leaching and losses of nitrate ($NO_3^-$) below the rooting zone (Aber et al., 1989; Perakis and Hedin, 2002). Continuing mineralization of soil organic N pools has been pointed out as the most probable reason for high N fluxes. Yet, the fate of excess $NO_3^-$ is not only controlled by belowground biogeochemical N-cycling and remineralization processes, but also by site-specific characteristics such as the size and quality of subsoil carbon pools, bedrock lithology, differential weathering and hydrological conditions (Lovett and Goodale, 2011). Altogether these parameters affect soil solution chemistries to produce complex spatial and temporal trends at a catchment scale.

Given the number of interacting factors affecting soil solution chemistries, there is an intrinsic difficulty associated to interpreting soil solution datasets. However, the chemical composition of soil solutions has been proven useful to assess the mobility of anionic species and nutrients in soils and their leaching from the soil profiles (Nieminen et al., 2016). It is thought

reflective of equilibrium between atmospheric deposition and soil physicochemical processes, including mineral weathering, sorption-desorption and cation exchange, as well as biological processes such as remineralization and nutrient turnover (Smith, 1976; Scott and Rothstein, 2014). In consequence, soil solution chemistries are increasingly seen as valuable indicators of perturbed forest ecosystems (Nieminen et al. 2016; Verstraeten et al., 2017).

Here, we interpret temporal and spatial relations between environmentally relevant chemical species in soil solutions collected using nests of lysimeters in a small, central European high-elevation catchment—U dvou loucek (UDL)—that formerly received high loads of atmospheric pollutants which resulted in soil acidification and spruce die-back. Some 30 years after peak acidification, the soil solutions at UDL were collected across an elevation gradient during a 2-year evaluation period. We revisited the unpublished chemical records to: (i) evaluate how do they reflect concurrent atmospheric deposition trends and

stream water fluxes of acidifying species and base cations (after Oulehle et al., 2017), and to what extent seasonal shifts observed on atmospheric deposition trends affected the spatially and temporally variable base cation contents of the independently measured soil solution chemistries; (ii) determine the effects of variable base cation content of soil solutions, soil granulometry and aluminum contents over the belowground carbon (C) and phosphorus (P) allocation; and (iii) assess to what extent does the chemistry of soil solutions varies along the topographic gradient. Our small catchment has a low

pedodiversity as it is situated entirely on base-poor gneissic bedrock in the north-eastern part of Czech Republic (Fig. 1a). This peculiarity simplifies our interpretations (*cf.* Kram et al., 2012). In addition, the contribution of groundwater *vs.* runoff infiltration is further evaluated by mean of a supplementary isotopic runoff model, which provides evidence for a likely contribution of groundwater enriched in selected chemical species due to sufficiently long water-saprolite interactions. Amongst 14 multi-decadal monitored small forested catchments of the Czech' GEOMON network, UDL received the highest

bulk atmospheric loads of a nitrogen and sulfur. As a result, the catchment is P limited and purportedly N saturated, with the ongoing pollution recovery process apparently altering concentrations and surface fluxes of other solutes via runoff (Oulehle et al., 2017). This paper addresses primarily soil solutions chemistry in the UDL catchment. Supporting data on the chemistry of spruce canopy throughfall and stream runoff—parameters which are used here for comparison purposes, are accessible in Oulehle et al. (2017). The combined dataset documents spatial heterogeneity of soil solutions in the form of variable nutrient

imbalances and offers further information to improve interpretations on the dynamics and catchment-scale patterns of soil solutions in temperate forests undergoing recovery after peak acidification.

**2 Study Site and Background Information**

This study was conducted in the UDL catchment, NE Bohemia, Czech Republic. Located in the Eagle Mountains (Orlické hory) at coordinates 50°13′ N, 16°29′ E (Fig. 1a), UDL is a 33-ha, V-shaped valley with a medium-gradient sloping land (Fig.

1b) incised within orthogneiss ($SiO_2$ = 75 wt. %; $Na_2O + K_2O$ = 8 wt. %; $MgO + CaO < 0.5$ wt. %). This acid metamorphic lithology comprises the Orlica-Snieznik Massif of Cambro-Ordovician age, together with blue schists of Neoproterozoic sedimentary protoliths that were intruded by the granitic protolith (Winchester et al., 2002; Don et al., 2003). A detailed description of the soils occurring on this watershed has been previously reported (Novak et al. 2005; Oulehle et al. 2017).

Accordingly, the soil in the catchment are mostly acidic Podzols developed on orthogneiss to which only the Entic qualifier applies. Low base status soils have developed at expense of the mineralogy of the orthogneissic bedrock, and given the lack of lithological discontinuities, pedodiversity is low across the catchment area with Mor being the most common humus.

With an elevation of 880-950 m a.s.l., UDL's climate is characterized as humid temperate. The mean precipitation is 1500 mm yr$^{-1}$, and the mean annual air temperature is 5 °C. An ephemeral snow cover lasts from late November to late March, when the monthly highest stream water runoff flow is usually recorded (~162 ± 29 mm). Vegetation cover complexity is low and essentially consist in Norway spruce (*Picea abies*, L. Karst). Following reforestation, UDL's vegetation cover includes approximately 27% of trees < 40 yrs in age, with 1.7 out of 33 ha remaining non-forested.

Historically, the site was influenced by emissions from nearby large industrial complexes. From the early 1970s, high anthropogenic discharges of $SO_2$ and $NO_x$ produced $H_2SO_4$, $HNO_3$ that affected the temperate forest ecosystem via wet and dry deposition. The largest point sources of these compounds were coal-burning power plants (Blazkova, 1996; Kolar et al., 2015). In central Europe alone, acid rain killed spruce stands over an area of approximately 1000 km$^2$ in the so-called "Black Triangle" (Novak et al. 2005). Emissions of acidifying compounds in these centrally planned economies peaked in the late 1980s; installation of desulfurization units in coal-burning power plants started in 1987 and was completed in the mid-1990s in both the Czech Republic and Germany, and several years later in southern Poland (Alewell et al., 2001; Hruska and Kram, 2003; Novak et al., 2005).

As in other coniferous forest ecosystems negatively affected by acid rain, in the Black Triangle area the productivity of temperate forests was likely perturbed by (i) enhanced leaching of base cations, such as potassium ($K^+$), calcium ($Ca^{2+}$) and magnesium ($Mg^{2+}$) (e.g., Gundersen et al., 2006;), and (ii) decreased bioavailability of phosphorous (P) (Gradowski and Thomas, 2008). UDL is also one of only three monitored catchments in the Czech Republic at elevations > 700 m a.s.l. whose forests were also affected between approximately 1975 and 1996 by massive acidification-related spruce dieback. After spruce defoliation, liming by aircraft was performed three times to raise the soil pH. Liming took place in 1986, 2002 and 2007, introducing three metric tons of ground dolomitic limestone per hectare into the mountain ecosystem on each occasion (Hruska pers. comm, 2018). Accordingly, during the decade 1994-2014, the median pH in the stream water remained stable in the range 5.2 ± 0.4, while over the same period, median pH levels measured in water percolating through the canopy (*i.e.,* throughfall) increased from 4.1 to 5.2 (Oulehle et al., 2017).

Historical input-output hydrochemical data are summarized in Table 1, and time-series concentration data for base cations ($Ca^{2+}$, $Mg^{2+}$, $K^+$, $Na^+$), nitrate ($NO_3^-$) and sulfate ($SO_4^{2-}$) are shown in Figure 2. According to Oulehle et al. (2017), UDL stream's pH was consistently acidic (< 5.5) during the studied period. For most elements (except for $Na^+$), the highest concentrations were observed in spruce canopy throughfall, followed by stream water ($SO_4^{2-}$, $Ca^{2+}$, $Mg^{2+}$, $K^+$) and open-area precipitation ($NO_3^-$). The median (1994-2014) sulfur (S) bulk atmospheric input was ~1.6 g m$^{-2}$ year$^{-1}$, which is far in excess of the atmospheric inputs observed in the remaining 13 GEOMON's monitored catchments across the Czech Republic (0.75 g m$^{-2}$ year$^{-1}$). Dissolved inorganic nitrogen (DIN) deposition input was 11.7 g m$^{-2}$ year$^{-1}$, thus exceeding the value observed at other monitored sites. The $Ca^{2+}$ input of 2.5 g m$^{-2}$ year$^{-1}$ largely exceeded the average $Ca^{2+}$ input into other monitored

catchments (0.6 g m$^{-2}$ year$^{-1}$). The Mg$^{2+}$ catchment input into UDL was 0.3 g m$^{-2}$ year$^{-1}$ (the median for all sites of the monitoring network was 0.1 g m$^{-2}$ year$^{-1}$). Inputs of Na$^+$ and K$^+$ were 0.6 and 1.3 g m$^{-2}$ year$^{-1}$, respectively (averages across GEOMON' sites were 0.2 and 0.5 g m$^{-2}$ year$^{-1}$).

## 3 Materials and Methods

### 3.1 Soil solution samples

In October 2010, five nests of Prenart suction lysimeters were installed at 50 cm depth below soil surface in a V-shaped arrangement as follow: hilltop west, hilltop east, slope west, slope east, and valley (filled circles in Fig. 1b). Each nest consisted of 3 lysimeters, and thus produced equal number of monthly replicates per sampling location. The lysimeter distributions along the V-shaped Shale Hills Critical Zone Observatory (Pennsylvania, USA; see Brantley et al., 2018 for a review) inspired our sampling design at UDL. The lysimeters in each nest were separated 6 to 10 m apart. The lysimeters were pressurized at the beginning of each sampling period using an electrical vacuum pump to 750 Bar below the atmospheric pressure at the time of sampling. During the first 12 months, soil solutions were collected each month and discarded. Following equilibration, the soil solutions were collected in 2 L PE bottles placed in a shallow soil pit. Monthly hydrochemical sampling of the lysimeters was performed during the following two hydrological years, *i.e.,* from November 2011 to October 2013. The collected soil solutions included both capillary and water percolating the mineral soil (*e.g.,* Nieminen et al., 2011).

### 3.2 Soil Samples

Five 0.5 m$^2$ soil pits were excavated in July 2015 at some distance to the previously installed suction lysimeter nests to avoid disturbances to the zero tension soil solution collection systems (Fig. 1b) while preserving a soil profile equivalent to the one at the nearby nest and also the relative position within the catchment area. The pits were excavated and processed by using the methodology described by Huntington et al. (1988) along both slopes of the UDL catchment (open circles in Figure 1b). Forest floor and mineral soil were removed to a depth of ~80 cm below surface and separated into topsoil in which plants have their roots, and which is comprised of the Oi/(L) + Oe(F) and Oa(H) as well as the top soil layers defined by depth (0-10, 10-20 cm) (not investigated here); and subsoil comprised by the 20-40 and 40-80 cm mineral soil layers. The soil layers were weighed in the field, processed by sieving to stones; coarse roots, and the > 1 cm soil fraction. Two kg of the < 1 cm soil fraction were transported to the laboratory. Only results from the 40-80 cm soil level are reported in this work. This level is considered in chemical equilibrium with waters collected by our 50 cm depth lysimeter nets and corresponds to horizon Bs2 in all plots.

### 3.3 Bulk Atmospheric Deposition and Stream Water Samples

Atmospheric deposition was sampled in open areas ("rainfall"), with sampling sites being 20 m apart among them and 1.2 m above ground (open square in Fig. 1b). Cumulative rainfall was collected monthly in three replicates. For oxygen isotope analyse, diffusive and evaporative losses from narrow-mouth bulk rain collectors were avoided by keeping precipitation under

a 5-mm layer of chemically stable mineral oil. Stream water samples and runoff flux estimations were collected monthly at a V-notch weir in the limnigraph location (Fig. 1b) according to methods outlined in Kram et al. (2003).

### 3.4 Analyses

#### 3.4.1 Soil Characterization

A Radiometer TTT-85 pH meter with a combination electrode was used to measure $pH_{H2O}$ of soil (soil–water suspension ratio = 1: 2.5). Soil texture was analyzed by the hydrometer method (ISO 11277 2009). Following air drying, the mineral substrate was sieved through a 2-mm sieve. The sieved samples were kept at 5ºC before chemical analysis. Exchangeable cations in soils were analyzed in 0.1 M $BaCl_2$ extracts by atomic absorption spectrophotometry (AAS, AAnalyst Perkin Elmer 200). Exchangeable acidity was determined by titration of the extracts. Cation exchange capacity (CEC) was calculated as the sum

of exchangeable base cations and exchangeable acidity. Base saturation (BS) was determined as the fraction of CEC associated with base cations. The concentrations of $NO_3^-$ and $SO_4^{2-}$ from the soil extracts described above were determined by ion chromatography (HPLC Knauer 1000), with limit of quantification (D.L.) of 0.1 and 0.3 mg $L^{-1}$, respectively. For an estimation of phosphorus (P) release, ammonium oxalate extractions were performed by following the protocol described in Schoumans (2000). In short, a reagent solution consisting of $(COONH_4)_2 \cdot H_2O$ and $(COOH)_2 \cdot 2H_2O$ was used to dissolve 1 g of the <2 mm

soil fraction. Extractable phosphorus ($P_{ox}$), iron ($Fe_{ox}$), and aluminium ($Al_{ox}$) were determined by shaking for 2 h in the dark duplicate samples of soils with 30 mL of 0.5 M acidified (pH 3.0) reagent in 50 mL centrifuge tubes. After shaking, centrifugation and filtration, the soil solutions were examined through inductively coupled plasma-atomic emission spectroscopy. The $P_{ox}$, $Fe_{ox}$, and $Al_{ox}$ concentrations were used to estimate the degree of P saturation of the soil [$DPS_{Ox} = P_{ox}*(0.5*(Fe_{ox} + Al_{ox})^{-1})$], which accounts for the P available to be released into solution in relation to the remaining binding

capacity of soil and, thus, allows identifying areas in the catchment with relatively higher potential for P export (Beauchemin and Simard, 1999; Borovec and Jan, 2018). For calculations of the amount of P sorbed by soil particles (Borovec and Jan, 2018), the average stream water P concentration, measured during our two years monitoring period (*i.e.*, 27.9 ± 6.5 µg $L^{-1}$), was used as an input for calculating the equilibrium P concentration in the catchment area.

#### 3.4.2 Soil Solutions

Concentrations of $NH_4^+$ and total phosphorus ($P_{tot}$) were measured spectrophotometrically (Perkin-Elmer Lambda 25; > 20 and 6 µg $L^{-1}$, respectively). Concentrations of $Na^+$, $K^+$, $Ca^{2+}$ and $Mg^{2+}$ were determined by electrothermal atomic absorption spectrometry (AAnalyst 200; > 5 µg $L^{-1}$). Concentrations of aluminium ($Al^{3+}$) were also measured by electrothermal atomic absorption instrument with a graphite furnace (D.L. < 0.01 mg $L^{-1}$). Concentrations of DOC and total dissolved nitrogen (TN) were determined on a combustion analyzer (Torch, Teledyne Temar; D.L. < 0.1 and 0.5 mg $L^{-1}$).

### 3.4.3 Statistical Analysis

The non-normally distributed (*i.e.,* non-parametric) data was evaluated by factor analysis. Empirical data were implemented in the computer code XLSTAT following the protocol by Vega et al. (1998). In short, data were normalized to zero and unit variance, and a covariance matrix of the normalized species was generated. For this analysis, the covariance matrix was diagonalized and the characteristic roots (eigenvalues) were obtained. The transformed variables, or principal components (PCs) so obtained were weighted linear combinations of the original plotted multidimensional variables. A rotation of PCs allowed a simpler and more meaningful representation of the underlying factors by decreasing the contribution of each variables to the two-dimensional plane. The variables can then be plotted in groups with correlation among them being determined by their position (*e.g.,* proximity, distance, orthogonality,). The two-dimensional plane where the rotated, normalized data were plotted can be interpreted in terms of the main controls over the general variance of the dataset (see Vega et al., 1998 for details).

### 4 Results

Table 2 lists physical data for mineral soil and chemical data for soil extracts from the 40-80 cm depth layer and data for soil solutions collected by suction lysimeters (50 cm depth). As described above, the dataset is grouped according to sampling position within the catchment area (*i.e.*, hilltops, slopes and valley; Fig. 1).

### 4.1. Soil Textural Characterization

In the eastern part of the catchment, coarse soil particles (gravel and stones) accounted for 24 % in the hillslope and 62 % of total soil granulometry at the hilltop, whereas in the western part of the catchment the soil particles above 10 cm in size accounted only for ~12 % (Table 2). The soil texture is loamy sand, with presence of authigenic clays (7-15 %) as weathering-induced alteration products of the orthogneiss parental material. The groundwater table is shallow.

### 4.2 Soil Chemical Characterization

### 4.2.1 pH, CEC and BS

The soil at the 40-80 cm depth was characterized by $pH_{H2O} < 5$ (Table 2). This median mineral soil layer had pH higher in the valley (4.7) compared to the hilltop (4. 4 to 4.2 units). The mean pH of soil solutions ranged similarly between the first and the second year, except for the valley (pH = 4.1 to 4.8; Table 2). The two-year averages of soil solutions were 5.2, 4.7, and 4.3 pH units on the hilltops, slopes, and valley, respectively. Therefore, the solid substrate extracts and soil solutions were characterized by an opposite elevational pH trend; *i.e.,* more acidic soil extracts uphill, more acidic soil solution downhill.

In the eastern part of the catchment, the average cation exchange capacity (CEC) of the mineral soil at 40-80 cm depth was up to 33 meq kg$^{-1}$ on the slope and 58 meq kg$^{-1}$ on the hilltop (Table 2). By contrast, in the western part, the CEC was 22 and 19

meq kg$^{-1}$, which is lower than the mean CEC values measured at all of the plots at UDL, whilst CEC in the valley was 27 meq kg$^{-1}$, which is within the mean CEC values measured at all of the plots at UDL: 32 ± 7 meq kg$^{-1}$ (Table 2). The range of base saturation (BS) values in the soil varied between 6 and 13 %, with higher BS observed in the east (> 9 %) as compared to the west (< 8 %). The CEC in the studied soil depth at UDL was dominated by exchangeable Al. Consequently, the soil base
saturation (BS) and soil pH$_{H2O}$ values were also low (Table 2). In summary, cation exchange capacity in UDL soils developed on base-poor orthogneiss ranged between 19 and 58 meq kg$^{-1}$ and base saturation was from 6 to 13 %. Both parameters had lower values than median values at the European LTER sites (84 meq kg$^{-1}$ and 30 %, respectively). With regard to other analogous central European forest ecosystems, soil solution solute concentrations in UDL were found above values also reported throughout the evaluation of temporal changes in inputs, runoff and fluxes (e.g., Manderscheid and Matzner, 1995a,b;
Wesselink et al., 1995, Hruska et al., 2000; Armbruster and Feger, 2004; Oulehle et al., 2006; Navratil et al., 2007).

BS at UDL was classified as poor with the dominant equivalent proportion of divalent base cations Ca$^{2+}$ (mean 46 %) and Mg$^{2+}$ (mean 24 %). The BS at UDL is twofold higher than the BS determined at similar soil depths in the leucogranitic catchment LYS (Kram et al., 1997; Hruska et al., 2000), which is the most acidified catchment of the Czech monitoring network (Oulehle et al., 2017). Holmberg et al. (2018) evaluated BS and CEC of numerous forest sites of the LTER (Long-
Term Ecological Research) network in nine European countries, with calculated median BS of 30 % and CEC of 84 meq kg$^{-1}$. From the European perspective, the soil BS and CEC values at the UDL were low.

### 4.2.2 Solute Concentrations

Mean concentrations of individual chemical species, such as DOC, sulfate, nitrate, base cations, aluminium, chloride and pH values in soil solutions collected at the 50 cm depth are listed separately for the years 2012 and 2013 (Table 1), whilst Figure
S1 shows their spatial variability of the statistical distribution (minimum, first quartile, median, third quartile and maximum; (in mg L$^{-1}$). Coefficients of variation within individual nests of lysimeters are listed in Table S1.

Concentrations of organic C (DOC) ranged between 0.40 and 1.81 wt. %, while concentrations of total nitrogen (TN) were between 0.02 and 0.10 wt. %, with [TN] being highest on hilltop east, and lowest on hilltop west (Table 2). Oxalate-extractable P was the lowest in the valley (334 mg kg$^{-1}$), and highest on hilltop east (536 mg kg$^{-1}$). The degree of P saturation varied
between 0.08 (valley) and 0.16 (hilltop east). These values fall below the lowest range observed in soil plots of the Czech Republic (see Borovec and Jan, 2018), pointing to a potential P deficit in the ecosystem.

Sulfate concentrations in soil solutions were on average 37 % lower than those in stream water, while relative to stream waters, NO$_3^-$ concentrations in soil solutions were 14 % lower. Similarly, the concentrations of K$^+$, Na$^+$, Ca$^{2+}$ and Mg$^{2+}$ were lower in soil solutions by 73, 63, 79 and 4 %, respectively. The combined dataset (Tables 1 and 2) show that except for DOC and Al$^{3+}$,
the rest of the studied chemical species were more diluted in the 50 cm' soil solutions than in the stream waters.

A time-series plot reveals that SO$_4^{2-}$concentrations in the valley were higher in winter than in summer (Figure 3). The mean SO$_4^{2-}$ concentrations in soil solution during the monitored period were found to be highest at the slopes (East > West), followed by the valley and hilltops (East ≈ West) (Table 2, Fig. S1). Our results for NO$_3^-$ across the lysimeter network also show that

this chemical species was readily bioavailable along the study site but mostly in the valley, where its concentrations were one order of magnitude higher than in the upslope soil solutions (Fig. S1). For this anion, the dataset also shows a high temporal variability, and in both years, $NO_3^-$ concentrations peaked by late spring in the valley (Fig. 3). By comparison, the belowground $NH_4^+$ concentrations were found to be low (usually below the limit of quantification, Table 2), a result that is consistent with previous observations at a soil research plot in north-western Czech Republic (Oulehle et al., 2006). For $SO_4^{2-}$ and $NO_3^-$, coefficients of variation were between 2 and 17 %, with no clear-cut differences within the sampling locations (Table S1).

Mean $Na^+$ and $K^+$ concentrations in soil solutions were the highest on slope east (Table 2; Fig. S1). For these cations, coefficients of variation (Table S1) were between 9 and 55 %, with the hilltop soil solutions exhibiting the largest variation in $K^+$ concentrations. The second year was characterized by generally lower $K^+$ concentrations in soil solutions collected in the valley, compared to the first year. $Na^+$ concentrations in soil solutions in the valley started to decrease only in the second half of the second year (Fig. 3).

As shown in Table 2, the highest mean $Mg^{2+}$ concentrations were observed on hilltop west, while the highest mean $Ca^{2+}$ concentrations were measured on slope east. The lowest mean $Mg^{2+}$ and $Ca^{2+}$ concentrations were found in the valley (Table 2, Figure S1). Coefficients of variations for $Mg^{2+}$ and $Ca^{2+}$ in soil solutions were relatively low, between 6 and 21 % (Table S1). The time series of $Ca^{2+}$ concentrations exhibited localized maxima in spring/early summer of the second year in soil solutions collected in most locations. Except for slope east, most locations also exhibited indistinct maxima in $Mg^{2+}$ concentrations in soil solutions in the spring to early summer of the second monitored year (Fig. 3).

### 4.3 Statistical Analysis

Several spatial trends are evident by evaluating the statistical distribution of anions and cations in the soil solutions (Table 2; Fig. S1). Throughout the monitored period there was a weak correlation between atmospheric deposition, stream water and soil solution concentrations (Fig. S2). The first factor of our explorative factor analysis, D1, exhibited a maximal overall variance that explained 24 % of total inter-correlated variance of collected data. The second factor, D2, had maximal variance amongst all unit length linear combinations that were uncorrelated to D1 and explained 18 % of variance within the dataset (Fig. S1). Based on the weights of the parameters, correspondence to each of these factors, and their cluster distribution, intrinsic properties of the soil, such as its DOC and clay contents (*i.e.*, D1) determined the variance on the soil solution solute concentration to a higher degree than seasonal inputs (i.e., D2).

### 5 Discussion

### 5.1 Comparison with Other European Forests

A comparison of previous studies with data presented here is not straightforward due to differences in sampling and analytical strategies, dissimilar and heterogeneous bedrock lithologies, variable soil buffering capacities, and other factors such as canopy density, inter-annual water influx variability and tree species diversity. Nonetheless, insights from previous related studies

provide the framework for our interpretations. Johnson et al. (2018) have recently published soil solution data from 162 plots monitored as part of the (ICP) Forest monitoring network, including median concentrations of environmentally relevant chemical species for the years 1998-2012. Soil solutions in the 40-80 cm deep mineral subsoil across Europe typically contained 6.3 mg $SO_4^{2-}$ $L^{-1}$, 1.0 mg $NO_3^-$ $L^{-1}$, 1.9 mg $Ca^{2+}$ $L^{-1}$, and 0.7 mg $Mg^{2+}$ $L^{-1}$ (*i.e.,* median values). Considering those

values alone, it follows that soil solutions at UDL in 2012-2013 were characterized by analogue concentrations of $SO_4^{2-}$, $Ca^{2+}$ and $Mg^{2+}$ as the ICP sites, and about one order of magnitude higher $NO_3^-$ concentrations than the ICP sites. Increased nitrate leaching toward the mineral soil in the UDL' forest ecosystems clearly reflects its N-saturated state (Aber et al., 1989; MacDonald et al., 2002).

## 5.2 Recovery from Acidification as Observed in the Spatial and Temporal Variability of Soil Solution Chemistries

In all studied soil solutions, a decreasing trend downhill was found in pH, DOC and $Ca^{2+}$ and $Mg^{2+}$ concentrations (Table 2; Figs. 3 and S1). Amongst such trends, those of pH—exhibiting a downhill 0.6 units difference, and DOC with concentrations lower by a factor of 2 to 3 in the valley as compared with hilltops, contrast with a rather moderate downslope decrease in soil solution' $Ca^{2+}$ and $Mg^{2+}$ concentration trends, which difference within the catchment area do not exceed 1 mg $L^{-1}$. Our time series data shows that $Ca^{2+}$ and $Mg^{2+}$ in soil solutions defined a general trend likely reflective of the balance between

evapotranspiration and biological inputs, with a punctual and correlative shift recorded in concentrations measured during mid-2013 (Table 2). This punctual correlative increase of these ions can be linked to an increase in strong anions inputs (Fig. 2), yet increased leaching of these macronutrients could also be regulated by temporary changes in the soil nitrate abundance (Oulehle et al., 2006; Wesselink et al., 1995; Akselsson et al., 2007, 2008). For DOC, the high variability on the slopes may reflect preferential flow paths. A relatively higher DOC belowground leaching on the eastern hilltop can be inferred from the

data, which suggests that C partitioning is site-specific, with little lateral redistribution from upslope organic soil levels toward the valley. Higher than topsoil mobilization of DOC below the rooted soil levels can be considered as a proxy for an incomplete recovery from acidification (Verstraeten et al., 2017).

Clear-cut seasonal concentration trends in soil solutions were recorded for $NO_3^-$ and $SO_4^{2-}$ (valley and slope west; Table 2). The underlying mechanism may be different for both anions. The co-evaluation of peak nitrate levels in soil solutions and

precipitation inputs during the monitoring period (Fig. S3) does not suggests a cause effect-relation linked to atmospheric deposition. Thus, higher abundance of $NO_3^-$ in soil solutions in the growing season may be related to higher rates of nitrification of organically cycled $NH_4^+$-N during summer (e.g., van Miegroet and Cole, 1984). Higher abundance of $SO_4^{2-}$ in soil solutions in winter remains unexplained. Historically, more soil S pollution was caused by higher $SO_x$ emissions from nearby coal-burning power plants during the cold season, but such seasonality was no longer seen for the years 2012-2013 (Fig. 2). Hence,

it is thought that nitrate in soil solutions during summer originated during the dormant season, whilst high sulfate concentrations observed during winter times originate from organic S being recycled and accumulated during the summer (Novak et al., 2001). As shown by Novak et al. (2005) using sulfur isotope ratios ($^{34}S$ : $^{32}S$), cycling of the high amounts of deposited $SO_x$ at UDL occurred not only by adsorption/desorption of $SO_4^{2-}$ onto soil particles, but also, to a great extent, by

cycling through the soil organic matter. As consequence acidification and export of $SO_4^2$, and for that matter, legacy reactive N, may prevail for several decades in the UDL and other similarly polluted mountain catchments (Novak et al., 2000; Armbruster et al., 2003; Mörth et al., 2005). Similarly, in UDL decreased runoff $NO_3^-$ export seems to be rather controlled by biological processes than by catchment hydrology (Oulehle et al., 2017). This is in association to increased organic

productivity, due to excess N, and because of the managed restoration of hardwood species following spruce die-back in UDL. Along with an increased total (aboveground) biomass immobilization of a large proportion of the atmospherically deposited $NO_3^-$ and $NH_4^+$ (e.g., McDowell et al., 2004), there was a concomitant increase in the demand of P, thus leading to an ecosystem deficiency on this macronutrient in the soil compartment.

Due to pollution abatement policies, atmospheric input has decreased since peak acidification, yet UDL has been previously

characterized by higher export of $SO_4^{2-}$, DOC, $Ca^{2+}$, $Mg^{2+}$, $K^+$, and $Na^+$ than their atmospheric input. In this regard, biogeochemical process within the soil seem to release more non-conservative ions than received from the atmosphere. Interestingly, export of total inorganic N from UDL *via* stream runoff continues to be significantly lower than its atmospheric input, but our results show that N leaching toward the subsoil levels is higher than runoff. Differences in porosity and greater fluid–rock-derived particle interactions, together with higher reactive surface areas and solute fluxes, might as well exert a

control over the measured soil solution chemical variability (Godsey et al., 2009). The latter effect seems to be critical control over the variability in soil solution chemistries at the hilltops, where the subsoil level contain significant amounts of coarse parental-rock material (Table 2).

For $Na^+$ and $K^+$ ions in soil solutions, our spatially resolved time-series observations (Fig. 3) show that their concentrations defined patterns and trends largely derived from heterogeneity in soil granulometry (Table 2), with seasonality and pulses in

atmospheric inputs also exerting some control over their concentrations is soil solutions (cf. Fig. 2 and Fig. 3). For $K^+$, and to a minor extent for $Na^+$, soil solution concentrations recorded peaks that are more or less correlative to $SO_4^{2-}$ and $NO_3^-$ inputs (cf., Figs. 2 and 3), again pointing to lapses in which the atmospheric contribution of strong anions exerted a significant control over the weathering and leaching of plagioclase and K-feldspar minerals in the bedrock (e.g., Moore et al., 2012). Oulehle et al. (2017) reported that $K^+$' average annual runoff was two to three times higher than that of $Na^+$ (Table 1), with both cations

exceeding runoff concentrations values measured in other monitored catchments. When the spatial variations in $Na^+$ and $K^+$ at the 50 cm depth soil solutions are also evaluated, a deep flow path within the eastern slope to the valley seems possibly augmented as a response of the solubilization of the $SO_4^{2-}$ atmospherically deposited and/or stored in the weathering zone below the rooted soils due to soil water saturation. Because $Na^+$ has low affinity toward organic and inorganic ligands in soil, and, thus, behaves relatively conservatively (McIntosh et al., 2017), a seemingly more rapid response of $Na^+$ than $K^+$ leaching

to soil solutions could be interpreted as the result of the episodic accumulation of strong anions belowground (cf. Fig. 2 and Fig. 3; *e.g.,* Spring 2013). From this result, it can be argue that localized and punctual chemical analyses of runoff waters in mountain catchments might not directly reflect nutrient partitioning trends along elevation gradients, but temporal variations of the strong anion content of the water table, which has implications for the design of studies centred in stream water analyses

for depicting the coupling of soil development processes and hydrology over variable time scales and between deep and shallow weathering processes.

The behaviour of $Na^+$ *vs.* $K^+$ ions can also be interpreted as a decrease in water residence time from the slope to the valley. On this note, we followed the modelling approach implemented by Buzek et al. (1995, 2009) to provide further insight on the

mean residence time of soil solutions—calculated across all sampling locations— which was estimated in approximately 8.3 months (Appendix A). In consequence, the runoff water at UDL is a mixture of direct precipitation with older soil solutions containing admixed with even older shallow groundwater. The supplementary isotopic modelling implemented here also shows that direct precipitation contributes between 20 and 40% of the discharge, with the rest being local soil pore and ground waters (Appendix A). The combination of all these three water types is called "mobile water", defined as the sum of all water pools

and fluxes that respond to changing precipitation amounts. This mobile water transiently increases soil solution saturation and concomitantly with such an increase, the hydrologic connectivity of soil pore waters to the stream can cause a heterogeneous distribution of dissolved ions in soil solutions at the catchment-scale (Basu et al., 2010).

Whilst factor analysis did not reveal significant relationships between measured UDL parameters (Fig. S2), cross-plots in Figure 4 show a relatively strong pH–Mg/Al correlation. Both variables in each cross-plot reached the highest values on slope

east and the lowest values in the valley. Correlations seem to follow a spatial trend determined by the higher solubility of Al bearing minerals at lower pH (Palmer et al., 2005). Finally, given the complexity of the possible interrelations among the environmental variables considered here, there was an apparent generally poor correlation between solute concentrations measured in the soil in 2012-2013 and decadal runoff and atmospheric deposition data compiled in Table 1 (after Ouhlele et al., 2017). Such a result in turn points to a major control exerted over the soil solution chemistry both by groundwater carrying

legacy pollutants and by spatially variable soil organic and inorganic ligand contents, which likely determine the residence time of each of the measured components.

## 5.3 Phosphorus Availability and Belowground C Allocation

Soil P sorption saturation is often used as an environmental indicator of soil P availability to runoff. Phosphorus losses from soils not subjected to an augmented erosional process are generally small (see Heuck and Spohn, 2016), with several factors

determining which fractions are transported in streams. Among them hill slopes, climate and soil type features are the most relevant determinants of the preferential transport of mainly fine-size fractions and associated elements, such as P typically associated to Al- and (to a minor extent in UDL) Fe-(oxyhydr)oxide fractions (Borovec and Jan, 2018, and references therein). Our calculation of P sorbed by the soil particles, as determined by oxalate extraction, shows that between 22 and 29 mg of P per kg of soil was sorbed in the in the 40-80 cm depth at the time of sampling, with insignificant difference between hilltops,

slopes and valley. It is possible that P limitation has developed because of the legacy of anthropogenic N deposition in this region. The homogenous pattern of low P availability contrasts with elevational differences in DOC concentrations in soil solutions (Fig. S1), which points to variable belowground leaching and allocation of C or could be reflective of variable inputs of C from regenerating vegetation in the N-saturated, P-limited forest ecosystem. Conifer tree species are generally more

tolerant to P limitations, which in turn make them more susceptible to nutrient depletion following losses from harvesting and exacerbated rates of nutrient export (Hume et al., 2018). We attribute the variable belowground allocation of C in UDL to spatially variable $Mg^{2+}$ and/or $K^+$ deficiencies (e.g., Rosenstock et al., 2016) rather than to P imbalances within the catchment since we detected no spatially contrasting P deficiencies exerting influence over the contrasting patterns of nutrient limitation and subsoil nutrients leaching observed across the studied forested landscape.

## 5 Conclusions

The hydrochemical comparisons implemented here were aimed at evaluating spatial and temporal concentration patterns on the water chemistry among the subsoil compartment of the critical zone in a temperate forest. Because of landscape and lithological simplicity, which facilitates discerning flow paths without variability effects introduced by differential bedrock weathering, it was possible discussing what factors in association to soil N-saturation affect the soil solution chemistries of a small mountainous catchment area reforested by Norway spruce after acidification-related spruce die-back. By combining soil solution chemical measurements and establishing comparisons with published hydrochemical data, this work provides evidence pointing to substrate variability, and C, but not P bioavailability, as major controls over the flux of base metal leached into the subsoil level and across the elevation gradient. Soil solutions at the 50 cm depth were generally more diluted than stream waters due to lateral surface runoff of solutions originating in the litter and humus enriched in $SO_4^{2-}$, $NO_3^-$, $K^+$, $Na^+$, $Ca^{2+}$ and $Mg^{2+}$. Increased concentrations are linked to anthropogenic atmosphere-derived pollution affecting natural (bio)geochemical processes. Differences between chemistry of soil solution and runoff could have been also caused by a direct contribution of throughfall, which scavenged atmospheric chemicals of the canopy and leached nutrients from inside the foliage, or by polluted open-area precipitation. Soil solutions had lower pH in the valley than at upslope locations, being more diluted in the valley than on hilltops in the case of DOC, $Ca^{2+}$ and $Mg^{2+}$. Both $NO_3^-$ and $SO_4^{2-}$ in soil solutions exhibited a clear seasonality that can affect base metal leaching, with maximum concentrations in the growing and dormant season, respectively. The observed temporal trends amongst strong anion inputs and leaching of base metals and acid anions reflect that at the time of sampling nutrient imbalances in UDL were linked to groundwater carrying legacy pollutants. A complementary isotope modelling show that the responses of the studied mountain catchment to precipitation are fast, *i.e.,* within the monthly sampling interval, with direct precipitation contributing 20 to 40% of the discharge and the rest being the contribution of local groundwater. When evaluated with regard to stream water chemistries and previously published input and fluxes, the dataset in this study provides insights into the localized controls and effects of acidification disturbances at a catchment-scale and offers a perspective of the spatially and temporarily variable nutrient concentrations in soil solutions that is relevant for (i) more effectively designing stream water chemical analyses aimed at understanding the coupling of soil development processes and hydrology over variable time scales, and between deep and shallow weathering processes in mountain catchments; and (ii) for evaluating soil recovery processes after atmospherically induced perturbations that affected other catchments analogue to UDL.

**Data availability.** The raw data can be requested from Daniel A. Petrash at the Czech Geological Survey.

**Supplement.** The following supplementary materials are available online at: ▆▆ , Figure S1: Descriptive statistics (2012-2013) for soil solution concentration values of dissolved organic carbon, sulfate, nitrate, base cations, aluminum and chloride (in mg L$^{-1}$) and pH values at 50 cm depth at UDL; Figure S2: Non-parametric multidimensional scaling ordination of time-series hydrochemical data for stream water, rainwater and soil solutions collected in lysimeters; Figure S3: Comparison of monthly precipitation volumes at UDL during the monitoring period (2012-2013) *vs.* the hydrologic years 2016-2017. Table S1: Coefficient of variation (Cv = $100\sigma/\mu$ ) of inorganic species across our lysimeter network.

**Author Contributions.** Conceptualization MN, FB, DP,; methodology MN, FB; data acquisition/validation JC, FV, BC, JC, TC, OM, FV, LB; visualization MS; formal analysis and investigation DP, PH; writing—original draft preparation, DP, MN, PH, PK; writing—review and editing, DP, PK, and MN.

**Competing interests.** The authors declare that they have no conflict of interest.

**Funding.** This research was funded by Czech Science Foundation (GACR), grant number 18-15498S.

**Acknowledgments.** We express our gratitude to Henning Meesenburg, Laure Soucémarianadin and one anonymous reviewer for valuable suggestions and thoughtful criticism that greatly improved the quality of this manuscript. Jakub Hruska and Tomas Navratil provided liming maps of the Eagle Mts. and data on oxalate extractions, respectively. We are grateful to Filip Oulehle for providing input-output hydrochemical data and his constructive criticism to an earlier version of the manuscript.

**Review statement.** This paper was edited by Teodoro Miano and reviewed by Henning Meesenburg and one anonymous reviewer.

## Appendix A

### A1 Hydraulic insights from $^{18}$O/$^{16}$O isotope modeling

Aiming at constraining the hydraulic parameters of the catchment under evaluation, a runoff generation model based on the water years 2016-2017, *i.e.*, on a later time period, was constructed as we believe it compares to the soil solutions during 2012-2013. To constrain the limitation of this approach, monthly precipitation among these periods were compared. As seen in Figure S3, annual precipitation measurements are comparable, with totals 1236, 1388, 1110, and 1284 mm in the hydrological years 2012, 2013, 2016 and 2017, respectively. Precipitation in the driest year 2016 corresponded to 80 % of precipitation in the wettest year 2013. Across this period, the mean monthly precipitation consistently peaked in December, May and September. Methodological details and mathematical components used to construct the isotopic $^{18}$O/$^{16}$O model are provided in the Appendix B (below).

Figure A1a shows that the $\delta^{18}O$ values of atmospheric input did not follow a canonical sinusoidal curve—isotopically heavy rainfall O in summer and isotopically light rainfall O in winter. Isotopically lighter $H_2O$-O in soil solutions relative to runoff in the spring of both years (Fig. A1b) indicate that water derived from the snowmelt predominates in soil pores several months toward summer. Isotopically heavier $H_2O$-O in soil solution, common in summer of the first year and in autumn of the second year, more closely corresponded to high $\delta^{18}O$ values of the instantaneous precipitation. Interestingly, $\delta^{18}O$ values of soil solutions in the valley (solid circles in Fig. A1) often departed from $\delta^{18}O$ values of runoff (thick solid line in Fig. A1b), despite the very small distance between the two sampling sites (70 m). Despite interpretative limitations imposed by different monitored periods, the runoff generation model can be generalized for the catchment interrogated here. Accordingly:

The response of the within-catchment hydrological system to precipitation is fast.

The hydrochemistry at the 5- cm soil depth reflected preceding precipitation events, modified by evapo-transpiration and, to a much smaller extent, mineral dissolution; the mixture mostly remained in soil pores until saturation was reached and leaching initiated; *cf.* Siegenthaler (1999).

The contribution of bulk precipitation to runoff is relatively low: 5 to 35 % (Fig. A1c).

The mean residence time of water in the UDL catchment (~8.3 months) was shorter than in three previously studied catchments in the Czech Republic. Lysina (LYS) catchment in the western Czech Republic (elevation of 830-950 m) was characterized by a mean water residence time of 15.2 months (Buzek et al. 2009). Dehtare and Jenin catchments in the central Czech Republic (elevations of 500-640 and 640-880 m) had a mean water residence time of 12.5 and 9.3 months, respectively (Buzek et al., 1995). A fourth small catchment located in a spruce die-back affected area near Jezeri (northwestern Czech Republic; elevation of 540-750 m) exhibited just slightly lower mean water residence time of 7.2 months than UDL (Maloszewski and Zuber 1982). While the bedrock at Jezeri and UDL was similar (gneiss), the steepness of both catchments differed (elevational span of 210 m at Jezeri *vs.* mere 70 m at UDL). The mean residence time of water at Jezeri and UDL was similar despite contrasting catchment areas (2.6 vs. 0.3 km$^2$).

**Appendix B**

**B1 O isotope analyses**

Atmospheric deposition was sampled in an open area ("rainfall"). Cumulative monthly rainfall was collected in three replicates, 20 m apart, 1.2 m above ground. Diffusive and evaporative losses from narrow-mouth rain collectors were avoided by keeping precipitation under a 5-mm layer of chemically stable mineral oil. Grab samples of runoff were collected monthly at the closing profile. The $\delta^{18}O_{H2O}$ values were obtained by off-axis integrated cavity output spectroscopy (OA-ICOS; Liquid Water Isotope Analyzer, Model 3000, LGR Inc., Mountain View, Ca, U.S.A.). One μL of water was injected through a port heated to 80°C. The vapor was transported into a pre-evacuated cavity and analyzed for the $^{18}O/^{16}O$ ratio. The reproducibility of $\delta^{18}O_{H2O}$ determinations was better than 0.20 ‰.

## B2 $\delta^{18}O_{H2O}$ modelling approach

A two-component model of runoff generation was produced using oxygen isotope ratios of $H_2O$ ($\delta^{18}O_{H2O}$) of open-area precipitation, runoff and suction lysimeters water. The model is derived from a general isotope mass balance calculated following Eq. (1):

$$\delta^{18}O_{tot} = \frac{\sum \delta^{18}O_i * Q_i}{Q_{tot}} \quad [\text{‰}],$$

(1)

where i is an individual water source, $Q_i$ is its mass flow [m³] and $Q_{tot}$ is the total flow [m³]. This mass balance is typically used for the separation of stormflow hydrograph into its event and pre-event components (Eq. (2)):

$$\delta_t Q_t = \delta_p Q_p + \delta_e Q_e \quad [\text{‰. m}^3 \text{ s}^{-1}],$$

(2)

where $Q_t$ is streamflow [m³.s⁻¹], $Q_p$ and $Q_e$ are contributing pre-event water (groundwater) and event water (rainfall, snowmelt) [m³.s⁻¹], and $\delta_t$, $\delta_p$ and $\delta_e$ are the corresponding isotopic compositions [‰]. Equation 2 can be solved parametrically for the contribution of the event water _p_ and of the pre-event water _(1-p)_ as shown in Eq. (3):

$$p = \frac{Q_e}{Q_t} = \frac{\delta_t - \delta_p}{\delta_e - \delta_p}.$$

(3)

The mass balance (1) is valid for any period of time if the isotope composition of all the components is known, for example for winter and summer. The mean annual $\delta^{18}O$ isotope composition (mean groundwater input), $\delta_{in}$, was estimated as the mean $\delta^{18}O_{tot}$ of the runoff.

A simple method of estimating the turnover time (mean age) of the subsurface reservoir employs an exponential model approximation; the distribution of transit times of water in the outflow is exponential and likely corresponds to permeability decreasing with the aquifer depth (Maloszewski and Zuber, 1982; Buzek, 1991). In case of stable isotopes, Siegenthaler (1979) demonstrated that the input (*i.e.,* precipitation) can be approximated by a sinusoidal function with a one-year period as per Eq (4):

$$\delta_{precip} = D + A \sin (2 \delta t),$$

(4)

where D = constant, A = amplitude of $\delta^{18}O$ variation in precipitation, t takes values 0-1 for a full-year period. Under a simplifying assumption of constant filtration and discharge, this input appears in discharge from the system as approximated by the factor B/A (Eq(5)):

$$\delta_{discharge} = D + B \sin (2 \delta t + \delta),$$

(5)

where B is the amplitude of $\delta^{18}O$ variation in output (discharge) a $\delta$ is the time shift of output variations in relation to input. The mean transit time (T) in years can be determined using Eq. (6) either the damping factor B/A or the phase shift $\delta$:

$$T = 1/2 \; \delta \; ((B/A)^{-2} -1)^{1/2}. \hspace{3cm} (6)$$

A similar approach can be applied also to lysimeters; $\delta_{precip}$ represents the input, and infiltrated soil solution ($\delta_{inf}$) is used instead of $\delta_{discharge}$.

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

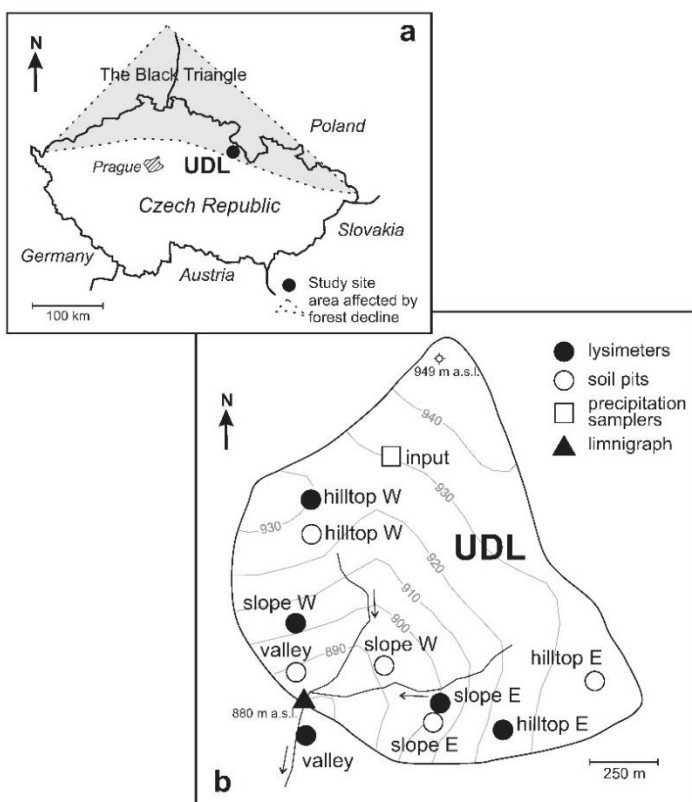

**Figure 1.** Study site location, **(a)** the shaded area shows the so-called "Black Triangle". **(b)** Sampling setup.

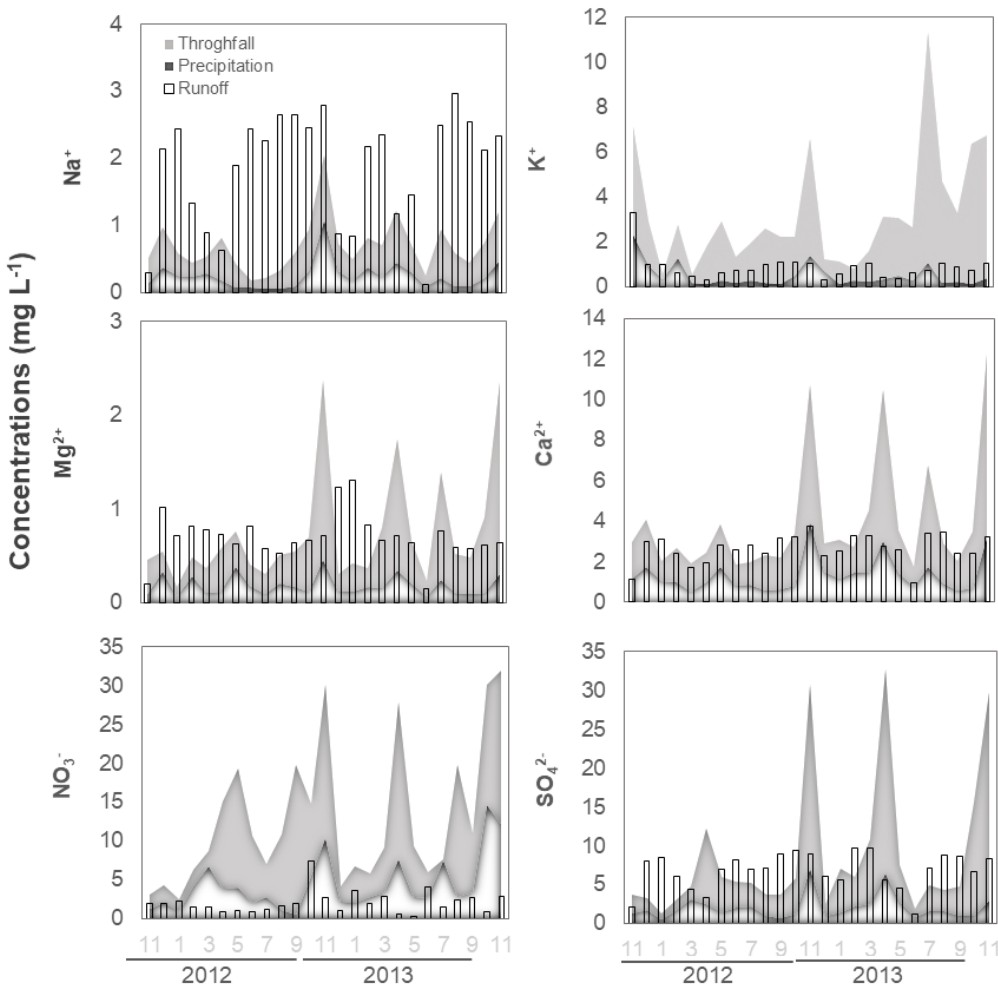

**Figure 2.** Hydrochemical data relevant for the monitoring period (2012-2013). X-axis shows months and hydrological year; concentrations after Oulehle et al. (2017).

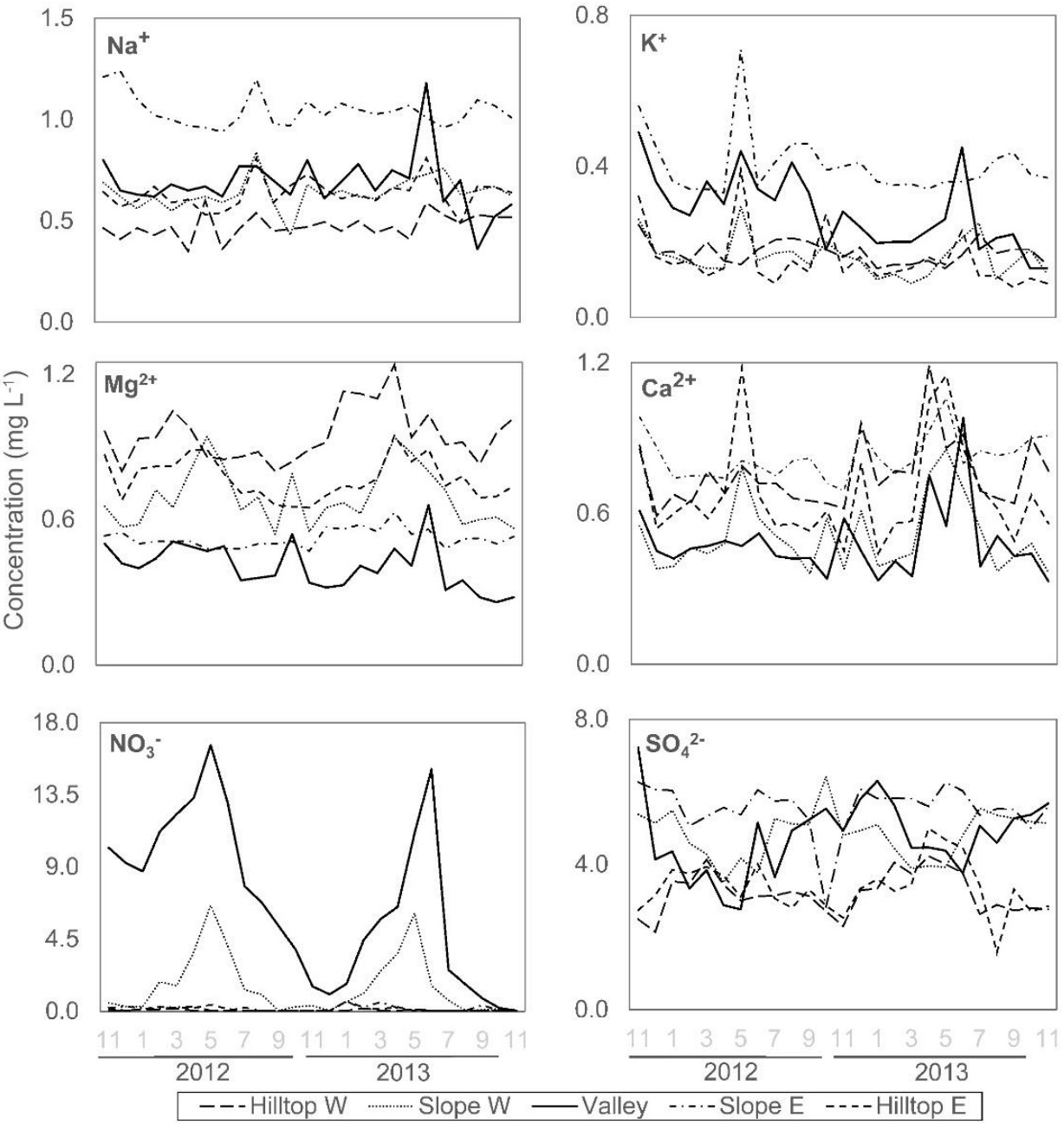

**Figure 3.** Spatially resolved, time-series soil solution concentration values of base cations, sulfate and nitrate at 50-cm depth. X-axis shows months and hydrological year**.**

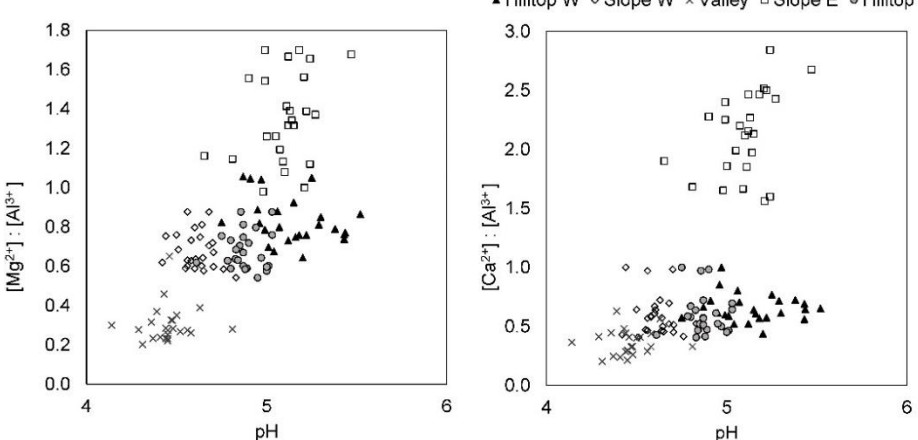

**Figure 4.** Comparison of Ca/Al and Mg/Al *vs.* pH.

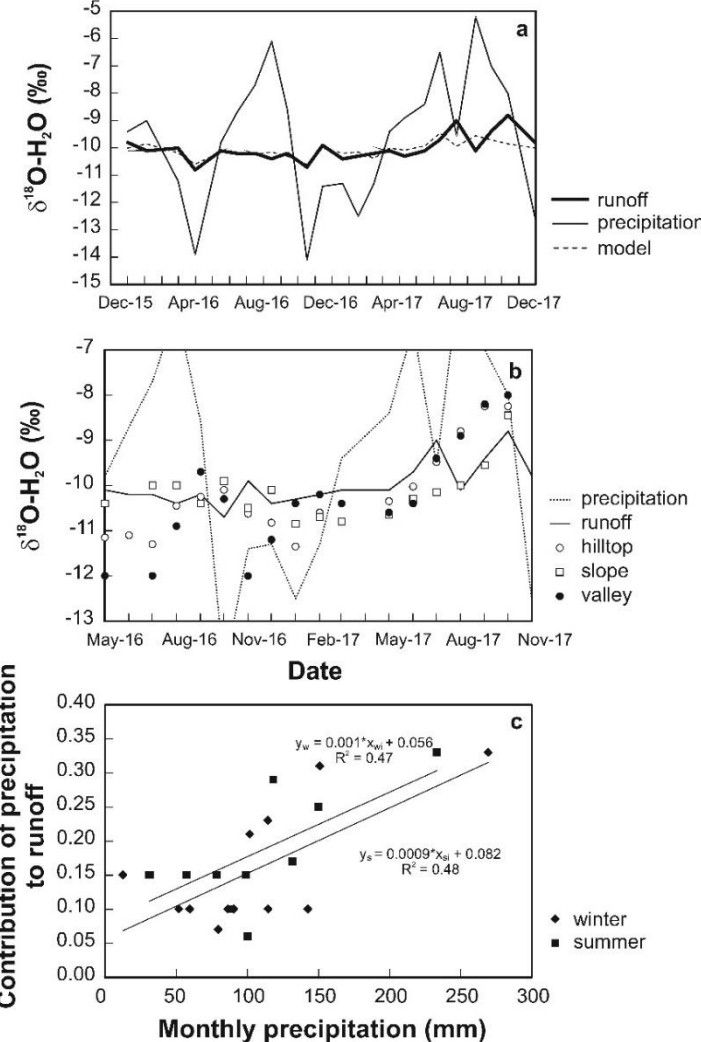

**Figure A1:** Time series of $\delta^{18}O$: (a) input-output model. (b) Areal distribution across the UDL catchment. (c) Estimated contribution of precipitation in runoff.

**Table 1.** Average hydrochemical data 2012-2013 (after Oulehle et al., 2017)

| | pH | $SO_4^{2-}$ | $NO_3^-$ | $NH_4^+$ | DOC | $Na^+$ | $K^+$ | $Mg^{2+}$ | $Ca^{2+}$ | $Al^{3+}$ | TP |
|---|---|---|---|---|---|---|---|---|---|---|---|
| | | | | | µg $L^{-1}$ | | | | | g $L^{-1}$ | |
| Rainfall | 5.7 | 2000 | 4400 | 700 | 2200 | 250 | 500 | 180 | 1350 | NM | < 20 |
| Throughfall | 5.5 | 6600 | 8500 | 1950 | 7900 | 450 | 2760 | 550 | 2600 | NM | < 20 |
| Runoff[#] | 5.9 | 6840 | 2070 | 40 | 7240 | 1850 | 840 | 700 | 2660 | 293 | 28 |
| **Standard Error (σ/√25)** | | | | | | | | | | | |
| Rainfall | 0.2 | 300 | 720 | 170 | 250 | 40 | 110 | 20 | 170 | - | - |
| Throughfall | 0.2 | 1550 | 1350 | 370 | 1400 | 40 | 470 | 100 | 430 | - | - |
| Runoff[#] | 0.1 | 460 | 290 | 10 | 850 | 170 | 110 | 50 | 140 | 37 | 6 |

[#]Average runoff flux during the monitoring period was 9.4 L s$^{-1}$, with maximums recorded in April (76.9 ± 4.0 L·s$^{-1}$) and minimums in August (0.5 ± 0.1 L s$^{-1}$). NM: not measured

**Table 2.** Spatially resolved physical and geochemical data for solid substrate (40-80 cm depth) and annual average soil solution chemistries at the 50-cm depth

| Measurement | Hilltop W | Slope W | Valley | Slope E | Hilltop E |
|---|---|---|---|---|---|
| **Soil** | | | | | |
| CEC (meq kg$^{-1}$) | 19.4 | 22.6 | 27.2 | 33.4 | 58.4 |
| BS (%) | 7.5 | 6.4 | 7.7 | 9.2 | 12.5 |
| >10 cm (kg ha$^{-1}$) ·10$^{3\ddagger}$ | 0 | 75 | 0 | 141 | 2038 |
| < 2-mm (kg ha$^{-1}$)·10$^{3\ddagger}$ | 4707 | 2842 | 2199 | 3726 | 1102 |
| pH$_{H2O}$ | 4.2 | 4.6 | 4.7 | 4.7 | 4.4 |
| Na$^+$ (mg kg$^{-1}$) | 3 | 6 | 17 | 6 | 34 |
| K$^+$ (mg kg$^{-1}$) | 30 | 19 | 4 | 29 | 7 |
| Mg$^{2+}$ (mg kg$^{-1}$) | 2 | 8 | 4 | 9 | 27 |
| Ca$^{2+}$ (mg kg$^{-1}$) | 7 | 27 | 20 | 26 | 78 |
| C$_{org}$ (%) | 0.40 | 0.81 | 0.99 | 0.45 | 1.81 |
| TN (%) | 0.020 | 0.037 | 0.045 | 0.032 | 0.101 |
| Al$_{Ox}$ (mg kg$^{-1}$) | 3880 | 5490 | 4390 | 2550 | 2370 |
| Fe$_{Ox}$ (mg kg$^{-1}$) | 1040 | 2500 | 3950 | 2810 | 4150 |
| P$_{Ox}$ (mg kg$^{-1}$) | 352 | 421 | 334 | 450 | 536 |
| DPS$_{Ox}$$^{\#}$ | 0.14 | 0.10 | 0.08 | 0.17 | 0.16 |

| Measurement | 2012 | 2013 | 2012 | 2013 | 2012 | 2013 | 2012 | 2013 | 2012 | 2013 |
|---|---|---|---|---|---|---|---|---|---|---|
| **Soil solution*** | | | | | | | | | | |
| pH | 5.4 | 5.5 | 4.6 | 4.4 | 4.1 | 4.5 | 5.0 | 4.8 | 4.9 | 4.9 |
| SO$_4$$^{2-}$ | 3132 | 3270 | 4850 | 4770 | 4420 | 5000 | 5440 | 5640 | 3360 | 3400 |
| NO$_3$$^-$ | 63 | 58 | 1800 | 1300 | 9870 | 4040 | 155 | 181 | 149 | 117 |
| NH$_4$$^+$ | < 20 | < 20 | < 20 | < 20 | < 20 | 30 | < 20 | < 20 | < 20 | 70 |
| DOC | 13500 | NA | 8440 | NA | 4510 | NA | 4230 | NA | 15100 | NA |
| Al$^{3+}$ | 1130 | 1170 | 945 | 859 | 1590 | 1280 | 396 | 394 | 1130 | 1150 |
| Na$^+$ | 455 | 493 | 611 | 662 | 683 | 687 | 1050 | 1040 | 618 | 649 |
| K$^+$ | 184 | 161 | 177 | 145 | 340 | 225 | 430 | 378 | 179 | 128 |
| Mg$^{2+}$ | 897 | 1000 | 700 | 699 | 445 | 378 | 505 | 539 | 775 | 764 |
| Ca$^{2+}$ | 699 | 806 | 498 | 531 | 459 | 514 | 794 | 851 | 668 | 697 |

$^\ddagger$ Soil particulate size. $^\#$Degree of P Saturation (DPS = P$_{ox}$·(0.5·(Fe$_{ox}$ + Al$_{ox}$)$^{-1}$). *Concentrations in µg L$^{-1}$. NA: not measured; variation coefficients are given in the supplementary material.

**FIGURE CAPTIONS**

**Figure 2:** Study site location: (a) The shaded area shows the so-called "Black Triangle"; (b) Sampling setup.

**Figure 2:** Hydrochemical data relevant for the monitoring period (2012-2013). X-axis shows months and hydrological year; concentrations after Oulehle et al. (2017).

**Figure 3:** Spatially resolved, time-series soil solution concentration values of base cations, sulfate and nitrate at 50-cm depth. X-axis shows months and hydrological year.

**Figure 4:** Comparison of Ca/Al and Mg/Al *vs.* pH.

**Figure A1:** Time series of $\delta^{18}O$: (a) input-output model; (b) Areal distribution across the UDL catchment; (c) Estimated contribution of precipitation in runoff.