# Peer review of "Spatially resolved soil solution chemistry in a central European atmospherically polluted high-elevation catchment"

_SOIL, 2019_

## Short Comment (SC1) · 28 Apr 2019

General comments

Petrash et al. present an interesting study on the spatial heterogeneity of soil solution in a central European high-elevation catchment which formerly received high loads of atmospheric pollutants. The topic is highly relevant for SOIL and the data gathered for the study are considered worth being published. However, with respect to the structure of the manuscript as well as the presentation of the methods and the data the manuscript seems to need major revisions. The introduction reflects the history of atmospheric pollution in the "Black Triangle" The last paragraph of the introduction

need to be completely rewritten as it contains little information concerning the design of the study, but methodological issues, results and even concluding statements. No objectives neither hypotheses are given in the introduction. Please amend accordingly. The methods section isn't detailed enough to be able to reproduce the approach completely. Information on the vacuum system of the lysimeters is missing. At least information on the applied pressure and if the vacuum was applied permanently or non-permanently should be given. Otherwise, assessments are mentioned (i.e. soil moisture determination), which value for the study remains unclear. According to Figure 1, sampling plots for soil solution and solid soil are up to more than 200 m apart. However, no information is given with respect to the comparability of the respective plots. Please give a rationale for this approach as the results from either solid soil or soil solution are related to the respective slope position. The contents of the results section are structured differently as the methods section. Some parameters, which are displayed in the tables and figures aren't mentioned in the text body of the results section. For soil solution, different units are reported in the text and in table 2. The last part of the results (i.e. page 8, row 6-11) seems to be more suitable for the discussion. The integration of the assessment of P availability into the study appears a little bit weak as no relation to either soil solution or runoff P concentrations is given. Also, a discussion on the role of P availability for tree nutrition is missing. Although discussed in the same sub-section (4.4), a convincing evidence for the "de-coupling" of P availability and organic carbon is missing. At least, an explanation should be given, why a "coupling" of P availability and organic carbon should be expected. The conclusion (5.) contains issues, that haven't been discussed before (e.g. the role of drought and torrential rains, isotope investigations). This should be avoided. The last three points of the conclusions resembles a collection of keywords more than elaborated findings. The chapter on 18O/16O modelling in the Appendix seems not very well integrated in the study. The references are mostly relevant for the study and up-to-date. However, some citations aren't very specific to the referenced issues. The publication of the manuscript can only be recommend after a major revision considering the

above mentioned concerns. Specific comments line-by-line are given in an attachment.

Please also note the supplement to this comment:
https://www.soil-discuss.net/soil-2019-9/soil-2019-9-SC1-supplement.pdf

**Supplement:**

Review of manuscript "Spatially resolved soil solution chemistry in a central European atmospherically polluted high-elevation catchment" by Daniel A. Petrash et al.

Specific comments
Note: "X -> Y" means replace "X" by "Y" in the text

| | |
|---|---|
| page 2, row 1 | "Blazkova et al., 1996" -> "Blazkova, 1996" |
| page 2, row 3 | "Blazkova et al., 1996" -> "Blazkova, 1996" |
| page 2, row 6 | "Fen et al. 1998" doesn't appear in the references. Is "Fenn et al. 1998" meant? If so, this citation doesn't support the before stated sentence, because it mainly reflects the situation in North America. |
| page 2, row 6 | "Hruska et al. 2003" -> "Hruska and Kram, 2003" |
| page 2, row 8 | "Gradowski et al., 2008" -> "Gradowski and Thomas, 2008" |
| page 2, row 9 | "Matschullat et al., 1998" -> "Matschullat et al., 1992". This citation doesn't give relevant information on the bioavailability of phosphorus. Please replace accordingly. |
| page 2, row 10-11 | This sentence isn't relevant for the introduction. It may be shifted to section 2.1. |
| page 2, row 10-18 | "For this aim, a nests of suction lysimeters was installed …" -> "For this aim, nests of suction lysimeters were installed …" |
| page 2, row 30 | Would this kind of orthogneiss with the reported composition really be termed "alkaline"? |
| page 3, row 1 | "… m, UDL's …" -> "… m a.s.l., UDL's …" |
| page 3, row 2 | At page 2, row 30, the parent material was named "alkaline gneiss". Is this the same as "porphyric granite"? Please clarify! How can Podzols develop on "alkaline gneisses"? |
| page 3, row 4 | Are the given figures the runoff for one month? Please clarify! |
| page 3, row 9 | I wouldn't consider saplings trees up to 40 yrs. |
| page 3, row 12 | Please delete "The." |
| page 3, row 12-13 | Please give the depth interval for the pH figures. |
| page 3, row 13 | "… a pH increases in throughfall measurements …" -> "… a pH increase in throughfall …". |
| page 3, row 13-14 | Two times the same citation in one sentence isn't necessary. |
| page 3, row 14-15 | A comparison of pH(H2O) and pH(KCl) doesn't makes much sense. |
| page 3, row 15 | "Hruska, 2000" -> "Hruska, pers. comm." (unpublished data weren't published in 2000). |
| page 3, row 16-26 | The content of this paragraph may be better placed in the results section. In addition figures presented should be given with standard deviation. Presented figures are a strange mixture of 13 or 14 catchments. Please revise consistently! |
| page 3, row 27 | Section "2.2.1" follows "2.1". Please check for consistency. |
| page 3, row 28 | "… at a 50-cm depth below …" -> "… at 50 cm depth below …".  Please change consistently throughout the manuscript. |
| page 3, row 28 | Were the lysimeters installed 50 cm below soil surface or below forest floor? Please clarify! |
| page 4, row 6 | The sampled depth intervals aren't horizons. Please specify! |
| page 4, row 8 | The given citation (FAO 2006) isn't about soil description. Please give correct citation! |
| page 4, row 14 | "open squares …" -> "open square …" |
| page 4, row 19 | Please specify the relation of soil to water for pH measurement. |

page 4, row 19-20    Please specify the meaning of soil moisture determination for the study, if any.

page 4, row 22    "5 ° C" -> "5°C"

page 4, row 23-27    The sequence of the determination of exchangeable cations seems in an incorrect order. Where $NO_3$ and $SO_4$ determined in the $BaCl_2$ extracts?

page 4, row 28    Consider "For a phosphorus (P) release estimation, …" -> "For a estimation of phosphorus (P) availability, …"

page 4, row 28-29    Please specify the soil:solution relation and the concentration of the solution.

page 5, row 1    "DPSox" in table 2.

page 5, row 3    "Beauchemin et al., 1999" -> "Beauchemin and Simard, 1999"

page 5, row 4    "Borovec et al., 2018" -> "Borovec and Jan, 2018"

page 5, row 11    Please introduce the meaning of D.L. at the first appearance! If detection limit is meant, consider to use "limit of quantification" instead.

page 5, row 14    Please explain the meaning of "non-parametric data".

page 5, row 14-22    The description of the factor analysis resembles - in my view as a non-statistician – a parametric method. Please specify the non-parametric component of the statistical analysis.

page 5, row 20    Please check the grammatical structure of the sentence.

page 5, row 24    The reporting of results for soil texture and pH in one subchapter appears strange to me. Why not report pH together with other soil chemical variables?

page 5, row 26-28    This sentence is more or less a repetition from section 2.2.1. Accordingly, it may be omitted.

page 6, row 1    Please give a definition of "pebbles" and "cobbles".

page 6, row 4-7    Consider to shift this part to section 3.3.

page 6, row 10    Is the mean given for all five sampling plots?

page 6, row 16    Please check "twice larger" for correctness.

page 6, row 17    "Hruska et al., 2001" isn't in the references. Please check.

page 7, row 4-6    Water volumes haven't been mentioned before. This sentence may be omitted or the relevance of water volumes for the study should be emphasized.

page 7, row 9    "… to be higher at …" -> "… to be highest at …"

page 8, row 2    According to figure S2, explained variance is 24%

page 8, row 3    According to figure S2, explained variance is 18%

page 8, row 3    Is "Fig. S2" meant here?

page 8, row 6-11    This part seems rather to be dedicated to the discussion. However, logical and grammatical consistency should be checked.

page 8, row 7    Please explain the meaning of "apparent insignificant correlation".

page 8, row 23-24    "Manderscheid et al., 1995" -> "Manderscheid and Matzner, 1995"

page 8, row 24    "Hruska et al., 2000" appears twice in the references. Which one is meant?

page 8, row 24    "Armbruster et al., 2004" -> "Armbruster and Feger, 2004"

page 9, row 9    "Meyer et al., 2001" is missing in references.

page 9, row 12-14    Please check grammatical consistency of the sentence.

page 9, row 20    Is "Novak et al., 2005" meant here?

page 10, row 4    What is meant with "areal control" here?

page 10, row 4-6    I can't follow this statement.

page 10, row 8-9    What is the rationale behind the comparison of K and Na outputs?

page 10, row 14-15    To which other period is the spring season of 2013 compared here?

page 10, row 21    "Heuck et al., 2016" -> "Heuck and Spohn, 2016"

page 10, row 23    "… in the 50 cm-depth …" -> "… in the 40-80 cm depth …"?

page 12, row 5     "… these periods ware compared." -> "… these periods were compared."

page 12, row 11   Figure A1a isn't visible in the manuscript. If "Figure 2A is meant, it should be corrected throughout Appendix A.

page 12, row 26   Consider "… in soil pore spaces …" -> "… in soil pores …"

page 12, row 28   What is meant with "direct precipitation"? If the "contribution of direct runoff (or "event water" as in eq. 2) to total runoff" is meant, a clear definition of "direct runoff" should be given.

page 15, row 21   "Fenn, E.M.;" -> "Fenn, M.E.;"

page 15, row 22   "Stottlemeye, R." -> "Stottlemyer, R."

page 15, row 24-25   "FAO: Guideline for soil description; Rome, Italy, 2006" should be cited here.

page 15, row 32-16/2 Please give correct title of the reference.

page 16, row 3-7 "Hruska et al., 2000" appears twice. Please indicate the respective citations with "a" and "b".

page 16, row 13   This line should be deleted.

page 16, row 22   "Ma, L; Teng, F-Z.; Lin, L.:" -> "Ma, L; Teng, F.-Z.; Lin, L.; et al.:"

page 18, row 2     "Soiling" -> " Solling"

page 19, Figure 1 (b)  Please consider to shift the sentence starting with "in the studied UDL …" to the text body of section 2.1.

page 23, Figure 2A   The content of this figure relates to Appendix A and should be placed in the supplements accordingly?

---

## Referee Comment (RC1) · Anonymous Referee #1 · 5 May 2019

In the paper, hydrological problems rather than soil properties are discussed. Unfortunately, the paper is not generally well written, organized and balanced. The introduction is very short and quite approximate. A real state of the art is completely missing. Introduction should more concisely lead to the objectives of the work. Usually, a general rationale is needed. What is the purpose of the study? Methods, results and discussion of soil properties are poor. Authors indicated that soil profiles were described according to FAO guidelines, but no descriptions are included in the paper. They investigated Podzols, but soil properties are described and discussed referring only to a layer of a depth of 40-80 cm, without taking into account soil genetic horizons. In two different sites on each hilltop and slope only one profile was investigated, and they

are compared with the properties of only one soil profile located in a valley. All together, five soil profiles were investigated, and soil properties were compared. Soils are very different in their properties, especially in a mountain areas, so the number of soil profiles was not sufficient to compare and to conclude on differences in such soil properties as carbon content, pH, cation exchangeable capacity, base saturation, etc. This, in addition to ignoring soil genetic horizons of Podzols investigated, is a reason of the weakness of data interpretation. Furthermore, soil properties, especially pH, changes in time through a year. It seems that soil samples were collected only once (in October 2010?). Why pH values are compared to pH values of water collected during a whole year (page 5 lines 25-26)? It is not clear what data were measured by authors and what data were cited from the literature (Oulehle et al. 2017). It is not clear how authors define runoff and throughfall (amount of water versus chemistry of water) and how these parameters were measured. These parameters should be defined. Conclusions summarize obtained results, but not contain real conclusions. Authors did not underline any innovative aspect that this article provides with respect to what is already present in literature. Several sentences are hard to follow, thus English proofreading is necessary. Summarizing, I do not consider this paper as relevant enough that deserves publication in the Soil journal. After a major revision it would be considered for publication in a journal dealing with hydrology.

**Detailed comments**

page 4, line 1: "A total of 15 replicates (3 per sampling location) were collected monthly". What replicates were collected? Does that mean water samples collected from 5 sites, each 3 times monthly?

page 4, lines 6-7: soil material from the depth 40-80 cm was collected. What soil horizons corresponded to this depth.

page 4, line 16: "Runoff samples were collected monthly at the limnigraph location". How these samples were collected?
page 4, line 4: "After centrifugation and filtration through 0.45 um cellulose–acetate filters, the filtrates were analyzed for cations" - this is not clear. Do these data refer to exchangeable cations? If so, the method was described in a wrong way. If not, what these data were measured for? Where these data were presented in the paper ?

page 5, lines 25-26: "Table 2 lists physical data for mineral soil and chemical data for soil extracts from the 40-80 cm depth layer and compares them with data for soil solutions collected by suction lysimeters (50-cm depth)" - what does mean "soil extracts"? Were they obtained as described on page 4, line 4? See comments to page 4, line 4.

page 6, line 3: "characterized by acidic pH". Reaction can be acidic, but pH may be high or low.

page 6, lines 4-5: " The mean pH of soil solutions ranged similarly between the first and the second year, except for the valley (pH valley of 4.1 in year 1, and 4.5 in year 2; Table 2). The two-year averages of soil solutions were", - this part of the text belongs to the paragraph 3.1. (Soil texture and pH). Does this data refer to soil extracts (page 4 line 4)? It seems that soil samples were collected just once in 2010, so how was it possible to obtain two-years averages? If these data refer to soil solutions obtained from lysimeters (page 3, line 28), they should be included in paragraph 3.3. (Solute concentrations in soil waters).

page 6, lines 9-10: how 33 meq and 58 meq can give mean 32 meq?

page 6, lines 10-17: usually cation exchange capacity differ significantly through Podzol profiles, from ectohumus through albic and spodic horizons. The same concerns base saturation. I have a doubt if these values may be compared without referring to given soil horizon.

page 6, line 26: authors used different terms to characterize chemical properties of water, namely: soil water, soil solution, mineral soil solution. It is not clear to what data refer concentration in soil water? Are they data from lysimeters? Nomenclature should

SOILD
be unified.

page 7, line 2: how was runoff measured? Were data in table 1 obtained by authors?

page 7, lines 10-11: " Our results for NO3- across the lysimeter network also show that this chemical species was readily bioavailable along mostly in the valley, where its concentrations were one order of magnitude higher than in the upslope soil solutions" - it is not clear how this was deduced.

page 10, line 3: table 2 does not provide sufficient information on soil textures heterogeneity

Figure 2 is unreadable due to unclear crosshatch. Does it present data obtained by authors?

Table 1. In Hydrology, throughfall is the process which describes how wet leaves shed excess water onto the ground surface. Was throughfall measured by authors? If so, how it was measured?

Table 2. Why (and how) soil particulate size (> 10 cm and

---

## Short Comment (SC2) · 6 May 2019

Soil is an intrinsic part of the Critical Zone (CZ), i.e., the thin, living skin of the earth, from the top of canopies down to saturated bedrock (Field et al., 2015). The bedrock in our mountain catchment exhibits no variation in lithology. Therefore, by combining soil solution chemical measurements and establishing meaningful comparisons with published hydrochemical data, we provide evidence pointing to carbon availability and landscape as major controls over the flux of water and solutes through the CZ in our small, N-saturated catchment (e.g., Chorover et al., 2011; Brantley et al., 2017). With our measurements and hydrochemical comparisons we aimed at evaluating spatial and

temporal alterations on the water chemistry among hydrological compartments of the CZ and calculated transit times within the soil. And because of landscape simplicity, which facilitates discerning flow paths, it is possible discussing potential variations in weathering products linked to N-saturation using our dataset.

Reviewer 1 kindly pointed out that the introduction in our original submission is deficient in highlighting the purpose of our study as briefly stated above. This flaw, which was also kindly noticed by Prof. Dr. Henning Meesenburg, is being carefully addressed in the revision of our MS.

In our view, understanding the coupling of soil development processes and hydrology over variable time scales, and between deep and shallow weathering processes remains as one of the major challenges of soil systems science. For this reason, we respectfully disagree with Reviewer 1 on the statement that the topic of our study does not concern to SOIL, but rather to a journal exclusively dealing with hydrology. We are addressing each of the reviewer's concerns and clarifying key aspects of our work aiming at publication in SOIL.

We thank the anonymous Reviewer 1 and Prof. Dr. Henning Meesenburg for their contributions to this SOIL Discussion.

References Brantley et al., 2017. Geomorphology, 277, 100–117. Chorover, J., et al., 2011. Vadose Zone J. 10, 884–899. Field, J. P., et al. 2015. Vadose Zone J., 14(1), 1–7.

———————————————

---

## Author Comment (AC3) · 1 Jul 2019

https://www.soil-discuss.net/soil-2019-9/soil-2019-9-AC2-supplement.pdf
* * *

---

## Author Comment (AC5) · 1 Jul 2019

Unmarked revised MS

Please also note the supplement to this comment:
https://www.soil-discuss.net/soil-2019-9/soil-2019-9-AC5-supplement.pdf
* * *

---

## Author Response (AR1)

Prague July 3, 2019

Prof. Dr. Teodoro Miano
Handling Topical Editor EGU SOIL
University of Bari Aldo Moro, Dip.to di Scienze del Suolo, della Pianta e degli Alimenti
Via G. Amendola, 165/A 70126 Bari, Italy

RE: Manuscript resubmission (ref. # soil-2019-9)

Dear Prof. Dr. Miano,

We thank the Editorial Board of SOIL for the opportunity to provide a revised version of our SOILD manuscript

15  titled "*Spatially resolved soil solution chemistry in a central European atmospherically polluted high-elevation catchment*". The revised version enclosed benefited significantly from the interactive review process, which allowed for progressively restructuring, clarifying and addressing flaws of the original paper. We highly value the reviewers' insights and criticisms and provide below point-by-point response to both of the reviews, together with a list of all relevant changes made in the manuscript and marked-up manuscript version. We express our

20  gratitude to the reviews for their insight that considerably improved our manuscript and your editorial handling on the topic.

On behalf of all authors,

25  Dr. Daniel Petrash
Czech Geological Survey
Geologicka 6 152 00 Prague 5 Czech Republic
daniel.petras@geology.cz; (+420) 774-143-577

**Reviewer 1**

**RC1.1** In the paper, hydrological problems rather than soil properties are discussed.

We have confidence that the "hydrological problems" considered in our *SOILD* are relevant for soil systems scientists focused on studying atmospheric-derived soil pollution and the detrimental propagation effects to contiguous ecosystems. See also response to RC1.3 below.

**RC1.2** Unfortunately, the paper is not generally well written, organized and balanced. The introduction is very short and quite approximate. A real state of the art is completely missing. Introduction should more concisely lead to the objectives of the work. Usually, a general rationale is needed. What is the purpose of the study?

We thank the reviewer for thoughtful criticism and kindly pointing out weakness of our original manuscript with regard to organization, balance, revision of pre-established concepts, and for carefully going through the text. A revised introduction addressing the reviewer's concerns was made available to the SOILD forum on Monday 04.05.19, with only minor editions made in the revised version of the manuscript. In addition, and as described below, by addressing each of the reviewer' concerns, requests for clarification and suggestions we have made every attempt to improve our report and look forward to the opportunity to submit the revised version of the SOILD.

**RC1.3** Methods, results and discussion of soil properties are poor. Authors indicated that soil profiles were described according to FAO guidelines, but no descriptions are included in the paper. They investigated Podzols, but soil properties are described and discussed referring only to a layer of a depth of 40-80 cm, without taking into account soil genetic horizons.

Descriptions of the soil profiles in UDL and their classification were published elsewhere (*e.g.,* Novak et al. 2005; Oulehle et al. 2017). This fact has been made clearer in the revised version of the manuscript. Constituents of the layer referred by the reviewer are necessarily mentioned since their mineralogy, granulometry (particulate surface area), variable organic content and weathering paths are considered to affect (or be reflected by) the spatial variability of the chemical composition of the subsoil pore waters collected via lysimeters in our small mountain catchment. With regard to soil texture, we have modified the manuscript as follow:

page 3, lines 32-33; 4 lines 1-4 (Study Site: Background Information): *"A detailed description of the soils occurring on this watershed has been previously reported (Novak et al. 2005; Oulehle et al. 2017). Accordingly, the soil in the catchment are mostly acidic Podzols developed on orthogneiss to which only the Entic qualifier applies. Low base status soils have developed at expense of the mineralogy of the orthogneissic bedrock, and given the lack of lithological discontinuities, pedodiversity is low across the catchment area with Mor being the most common humus."*

page 7, lines 17-20 (Results): *"In the eastern part of the catchment, coarse soil particles (gravel and stones) accounted for 24 % in the hillslope and 62 % of total soil granulometry at the hilltop, whereas in the western part of the catchment the soil particles above 10 cm in size accounted only for ~12 % (Table 2). The soil texture is loamy sand, with presence of authigenic clays (7-15 %) as weathering-induced alteration products of the orthogneiss parental material. […]"*

**RC1.4** In two different sites on each hilltop and slope only one profile was investigated, and they are compared with the properties of only one soil profile located in a valley. All together, five soil profiles were investigated, and soil properties were compared. Soils are very different in their properties, especially in a mountain areas, so the number of soil profiles was not sufficient to compare and to conclude on differences in such soil properties as

carbon content, pH, cation exchangeable capacity, base saturation, etc. This, in addition to ignoring soil genetic horizons of Podzols investigated, is a reason of the weakness of data interpretation.

We respectfully disagree with the reviewer in his/her assertion that one soil profile for sampling area is insufficient for discussing the likely effect of soil textural features on the variability of the parameters listed above *vs.* legacy pollutants and nutrients imbalances. Previous, more detailed soil profile analyses in UDL has consistently show low pedodiversity (see note above on this regard).

**RC1.5** Soil properties, especially pH, changes in time through a year. It seems that soil samples were collected only once (in October 2010?).

We thank the reviewer for carefully going through the paper and for requesting sound clarifications. The soil samples were obtained in July 2015. This is now mentioned in page 5, line 17. On the other hand, we intended to compare pH values measured in stream water *vs.* pH of soil solutions. The confusing typo has been corrected as follow (page 4, line 24-26): *"[…] during the decade 1994-2014, the median pH in the stream water remained stable in the range 5.2 ± 0.4, while over the same period, median pH levels measured in water percolating through the canopy (i.e., throughfall) increased from 4.1 to 5.2 (Oulehle et al., 2017)."*

**RC1.6** Why [soil] pH values are compared to pH values of water collected during a whole year (page 5 lines 25-26)?

In the revised text, the lines highlighted here now read (page 6, lines 13-16): *"Table 2 lists physical data for mineral soil and chemical data for soil extracts from the 40-80 cm depth layer and data for soil solutions collected by suction lysimeters (50 cm depth). As described above, the dataset is grouped according to sampling position within the catchment area (i.e., hilltops, slopes and valley; Fig. 1)."* **That is, for the pH parameter there is no reference to the table directly serving for comparison purposes between discrete samples.**

**RC1.7** It is not clear what data were measured by authors and what data were cited from the literature (Oulehle et al. 2017).

To make clearer this key aspect highlighted by the reviewer, we have restructured the manuscript. Accordingly, the revised introduction now includes the following lines (page 3, lines 22-24): *"This paper addresses primarily soil solutions chemistry in the UDL catchment. Supporting data on the chemistry of spruce canopy throughfall and stream runoff—parameters which are used here for comparison purposes, are accessible in Oulehle et al. (2017)."*

Also, the first call to pre-existing data used for comparison purposes in the text is made in the new section *2 Study Site and Background Information* (*i.e.,* Fig. 2 and Table 1)

**RC1.8** It is not clear how authors define runoff and throughfall (amount of water versus chemistry of water) and how these parameters were measured. These parameters should be defined.

Following this important reviewer's request for clarification, the word "runoff", when originally and unclearly referring to runoff water samples, has been substituted for "stream water(s)" and a brief definition of throughfall water is now provided in page 4, lines 25-26: *"[…] measured in water percolating through the canopy (i.e., throughfall) […]"*

**RC1.9** Conclusions summarize obtained results, but not contain real conclusions. Authors did not underline any innovative aspect that this article provides with respect to what is already present in literature.

The Conclusions section has been rewritten taking in consideration this concern of the reviewer and additional suggestions to improve the section provided by Reviewer 2.

**RC1.10** Several sentences are hard to follow, thus English proofreading is necessary.

Careful proofreading have been conducted to avoid runoff sentences.

**RC1.11** Summarizing, I do not consider this paper as relevant enough that deserves publication in the Soil journal.

We have made every attempt to address the reviewer's criticism and requests for clarification, elucidated its importance to the SOIL journal' readership in the Introduction and Discussion section, and, thus, hope that the revised version of our SOILD could be considered for publication in the EGU SOIL.

**RC1.12** After a major revision it would be considered for publication in a journal dealing with hydrology.

Please see to the author's reply to RC1.1 and RC1.11.

Detailed comments

**RC1.13** page 4, line 1: "A total of 15 replicates (3 per sampling location) were collected monthly". What replicates were collected? Does that mean water samples collected from 5 sites, each 3 times monthly?

This request of clarification of the reviewer has been addressed as follow (page 5, lines 7-8): *"[…] Each nest consisted of 3 lysimeters, and thus produced equal number of monthly replicates per sampling location. […]"*

**RC1.14** page 4, lines 6-7: soil material from the depth 40-80 cm was collected. What soil horizons corresponded to this depth

This request of clarification of the reviewer has been addressed as follow (page 5, lines 25-26): *"Only results from the 40-80 cm soil level are considered here. This level is in chemical equilibrium with waters collected by our 50 cm depth lysimeter nets and correspond to horizon Bs2 in all plots."*

**RC1.15** page 4, line 16: "Runoff samples were collected monthly at the limnigraph location". How these samples were collected?

To address this reviewer's query, the revised text now reads (page 6, lines 1-2): *"Stream water samples and runoff flux estimations were collected monthly at a V-notch weir in the limnigraph location (Fig. 1b) according to methods outlined in Kram et al. (2003)."*

**RC1.16** page 4, line 4: " After centrifugation and filtration through 0.45 um cellulose–acetate filters, the filtrates were analyzed for cations" - this is not clear. Do these data refer to exchangeable cations? If so, the method was described in a wrong way. If not, what these data were measured for? Where these data were presented in the paper?

To address this flaw of the methods description kindly pointed out by the reviewer, the revised text has been streamlined as follow (page 6, lines 7-11): *"Exchangeable cations in soils were analyzed in 0.1 M BaCl$_2$ extracts by atomic absorption spectrophotometry (AAS, AAnalyst Perkin Elmer 200). Exchangeable acidity was determined by titration of the extracts. Cation exchange capacity (CEC) was calculated as the sum of exchangeable base cations (Ca$^{2+}$, Mg$^{2+}$, K$^+$, Na$^+$) and exchangeable acidity. Base saturation (BS) was determined as the fraction of CEC associated with base cations"*

**RC1.17** page 5, lines 25-26: "Table 2 lists physical data for mineral soil and chemical data for soil extracts from the 40-80 cm depth layer and compares them with data for soil solutions collected by suction lysimeters (50-cm depth)" - what does mean "soil extracts"? Were they obtained as described on page 4, line 4?

Please see note above.

**RC1.18** page 6, line 3: "characterized by acidic pH". Reaction can be acidic, but pH may be high or low.

The offending line highlighted by the reviewer now reads (page 7, line 23): *"The soil at the 40-80 cm depth was characterized by $pH_{H2O} < 5$"*

**RC1.19** page 6, lines 4-5: "The mean pH of soil solutions ranged similarly between the first and the second year, except for the valley (pH valley of 4.1 in year 1, and 4.5 in year 2; Table 2). The two-year averages of soil solutions were", - this part of the text belongs to the paragraph 3.1. (Soil texture and pH). Does this data refer to soil extracts (page 4 line 4)? It seems that soil samples were collected just once in 2010, so how was it possible to obtain two-years averages? If these data refer to soil solutions obtained from lysimeters (page 3, line 28), they should be included in paragraph 3.3. (Solute concentrations in soil waters).

The data refer to soil water solutions obtained from lysimeters, not to soil extracts. As per reviewer's suggestion the text has been moved to subsection "4.2.1 pH, CEC and BS" of the restructured manuscript

**RC1.20** page 6, lines 9-10: how 33 meq and 58 meq can give mean 32 meq?

We thank the reviewer for highlighting the lack of clarity of the initial data description. The revised text now reads (page 8, line 2): *"[…] which is within the mean CEC values measured at all of the plots at UDL: 32 ± 7 meq kg$^{-1}$ (Table 2)."*

**RC1.21** page 6, lines 10-17: usually cation exchange capacity differ significantly through Podzol profiles, from ectohumus through albic and spodic horizons. The same concerns base saturation. I have a doubt if these values may be compared without referring to given soil horizon.

To address this important observation of the reviewer. The following feature was added to our description of Soil Samples (page 5, line 25-26): *"Only results from the 40-80 cm soil level are reported in this work. This level is considered in chemical equilibrium with waters collected by our 50 cm depth lysimeter nets and corresponds to horizon **Bs2** in all plots."*

**RC1.22** page 6, line 26: authors used different terms to characterize chemical properties of water, namely: soil water, soil solution, mineral soil solution. It is not clear to what data refer concentration in soil water? Are they data from lysimeters?

A unification of equivalent terms referring to soil water from lysimeters has been implemented in this revision, and are now only referred to as "soil solutions"

**RC1.22** page 7, line 2: how was runoff measured?

Please refer to author's answer to RC1.15 above

**RC1.23** Were data in table 1 obtained by authors?

Table 1 is comprised of background information and this has been now made clearer. Please see author's response to RC1.7 (above)

**RC1.24** page 7, lines 10-11: "Our results for NO3- across the lysimeter network also show that this chemical species was readily bioavailable along mostly in the valley, where its concentrations were one order of magnitude higher than in the upslope soil solutions" - it is not clear how this was deduced.

The revised text (page 8, 33 and 9, 1-2) now reads: *"Our results for $NO_3^-$ across the lysimeter network also show that this chemical species was readily bioavailable along the study site but mostly in the valley, where its concentrations were one order of magnitude higher than in the upslope soil solutions (__Fig. S1__)."*

**RC1.25** page 10, line 3: table 2 does not provide sufficient information on soil textures heterogeneity

Table 2 only presents the available relevant information on soil texture for comparison purposes in the studied low pedodiversity, UDL catchment area.

**RC1.26** Figure 2 is unreadable due to unclear crosshatch. Does it present data obtained by authors?

As per reviewer request the crosshatch has been removed from the revised version of Figure 2 thus making it clearer. Figure 2 is background information and in consequence is first called in section 2, *Study Site and Background Information*.

[Figure]

**RC1.27** Table 1. In Hydrology, throughfall is the process which describes how wet leaves shed excess water onto the ground surface. Was throughfall measured by authors? If so, how it was measured?

Generally accepted definition as used in our manuscript and related literature is the water percolating through the canopy (please see also author's response to RC1.8).

**RC1.28** Table 2. Why (and how) soil particulate size (> 10 cm) were expressed in t/ha?

From the methods of estimation outlined in section 3.2. Soil Samples, where the weighted material from a given area. To address this concert, this parameter in Table 2 is now expressed in kg instead of ton.

**RC1.29** Figure S2. What does mean the following: "hydrochemical data for runoff, atmospheric in lysimeters"?

We thank the reviewer for carefully going throughout the manuscript. The offending line caption now reads: [...] *"Non-parametric multidimensional scaling ordination of time-series hydrochemical data for stream water, rainwater and soil solutions collected in lysimeters."*

**Reviewer 2**

**RC2.1** Petrash et al. present an interesting study on the spatial heterogeneity of soil solution in a central European high-elevation catchment which formerly received high loads of atmospheric pollutants. The topic is highly relevant for SOIL and the data gathered for the study are considered worth being published. However, with respect to the structure of the manuscript as well as the presentation of the methods and the data the manuscript seems to need major revisions.

We thank the constructive criticism of reviewer 2, Prof. Dr. Meesenburg, and have made every attempt to address each of his concerns and suggestions for improvement.

**RC2.2** The introduction reflects the history of atmospheric pollution in the "Black Triangle" The last paragraph of the introduction needs to be completely rewritten as it contains little information concerning the design of the study, but methodological issues, results and even concluding statements. No objectives neither hypotheses are given in the introduction. Please amend accordingly.

To address this concern of the reviewer, the introduction section has been significantly rewritten and expanded to now provide a brief review of background information and relevance, objectives and hypotheses of the study.

**RC2.4** Otherwise, assessments are mentioned (i.e. soil moisture determination), which value for the study remains unclear.

Unused information such as unrelated soil moisture measurements has been now removed from the revised methods.

**RC2.5** According to Figure 1, sampling plots for soil solution and solid soil are up to more than 200 m apart. However, no information is given with respect to the comparability of the respective plots. Please give a rationale for this approach as the results from either solid soil or soil solution are related to the respective slope position.

The revised version of the MS page 5, rows 19-19 now clarifies this aspect: *"Five 0.5 m² soil pits were excavated in July 2015 at some distance to the previously installed suction lysimeter nests to avoid disturbances to the zero tension soil solution collection systems (Fig. 1b) while preserving a soil profile equivalent to the one at the nearby nest and also the relative position within the catchment area."*

**RC2.6** The contents of the results section are structured differently as the methods section.

The revised results section of our MS has been restructured to address this flaw of the original submission kindly pointed out by the reviewer.

**RC2.7** Some parameters, which are displayed in the tables and figures aren't mentioned in the text body of the results section.

Conductivity parameter has been removed from Table 2. The rest of the parameters listed are either mentioned/discussed in the main text, or are needed for calculations (*e.g.,* $Al_{Ox,}$ $Fe_{Ox}$)

**RC2.8** For soil solution, different units are reported in the text and in table 2.

Now all in-text mentions to soil solution concentrations are referred to Table 2, where concentrations units are expressed in ppb.

**RC2.9** The last part of the results (i.e. page 8, row 6-11) seems to be more suitable for the discussion.

The misplaced text kindly highlighted by the reviewer was moved and integrated to the discussion section (page 12, rows 16-21).

**RC2.10** The integration of the assessment of P availability into the study appears a little bit weak as no relation to either soil solution or runoff P concentrations is given. Also, a discussion on the role of P availability for tree nutrition is missing. A convincing evidence for the "de-coupling" of P availability and organic carbon is missing. At least, an explanation should be given, why a "coupling" of P availability and organic carbon should be expected.

There is no further reference in the revised text to coupled/decoupled cycling of nutrients, but to the role of P in belowground C allocation and base cation imbalance. With regard to our P measurements, the revised text (page 12, rows 23-34) now reads: *"Soil P sorption saturation is often used as an environmental indicator of soil P availability to runoff. Phosphorus losses from soils not subjected to an augmented erosional process are generally small (see Heuck and Spohn, 2016) […]. It is possible that P limitation has developed because of the legacy of anthropogenic N deposition in this region. The homogenous pattern of low P availability contrasts with elevational differences in DOC concentrations in soil solutions (Fig. S1), which points to variable belowground leaching and allocation of C or could be reflective of variable inputs of C from regenerating vegetation in the N-saturated, P-limited forest ecosystem. Conifer tree species are generally more tolerant to P limitations, which in turn make them more susceptible to nutrient depletion following losses from harvesting and exacerbated rates of nutrient export (Hume et al., 2018). We attribute the variable belowground allocation of C in UDL to spatially variable Mg2+ and/or K+ deficiencies (e.g., Rosenstock et al., 2016) rather than to P imbalances within the catchment since we detected no spatially contrasting P deficiencies exerting influence over the contrasting patterns of nutrient limitation and subsoil nutrients leaching observed across the studied forested landscape."*

**RC2.11** The conclusion contains issues, that haven't been discussed before (e.g. the role of drought and torrential rains, isotope investigations). This should be avoided.

The flaws on the original conclusions kindly pointed out by the reviewer are now avoided and the conclusion section have been streamlined, with no reference to undiscussed aspects.

**RC2.12** The last three points of the conclusions resembles a collection of keywords more than elaborated findings.

The last three points indicated by the reviewer were removed from the text, and the section rewritten as follows:

*"The hydrochemical comparisons implemented here were aimed at evaluating spatial and temporal concentration patterns on the water chemistry among the subsoil compartment of the critical zone in a temperate forest. Because of landscape and lithological simplicity, which facilitates discerning flow paths without variability effects introduced by differential bedrock weathering, it was possible discussing what factors in association to soil N-saturation affect the soil solution chemistries of a small mountainous catchment area reforested by Norway spruce after acidification-related spruce die-back. By combining soil solution chemical measurements and establishing comparisons with published hydrochemical data, this work provides evidence pointing to substrate variability, and C, but not P bioavailability, as major controls over the flux of base metal leached into the subsoil level and across the elevation gradient. Soil solutions at the 50 cm depth were generally more diluted than stream waters due to lateral surface runoff of solutions originating in the litter and humus enriched in SO42-, NO3-, K+, Na+, Ca2+ and Mg2+. Increased concentrations are linked to anthropogenic atmosphere-derived pollution affecting natural (bio)geochemical processes. Differences between chemistry of soil solution and runoff could have been also caused by a direct contribution of throughfall, which scavenged atmospheric chemicals of the canopy and leached nutrients from inside the foliage, or by polluted open-area precipitation. Soil solutions had lower pH in the valley than at upslope locations, being more diluted in the valley than on hilltops in the case*

*of DOC, Ca2+ and Mg2+. Both NO3- and SO42- in soil solutions exhibited a clear seasonality that can affect base metal leaching, with maximum concentrations in the growing and dormant season, respectively.*

*The observed temporal trends amongst strong anion inputs and leaching of base metals and acid anions reflect that at the time of sampling nutrient imbalances in UDL were linked to groundwater carrying legacy pollutants. A complementary isotope modelling show that the responses of the studied mountain catchment to precipitation are fast, i.e., within the monthly sampling interval, with direct precipitation contributing 20 to 40% of the discharge and the rest being the contribution of local groundwater. When evaluated with regard to stream water chemistries and previously published input and fluxes, the dataset in this study provides insights into the localized controls and effects of acidification disturbances at a catchment-scale and offers a perspective of the spatially and temporarily variable nutrient concentrations in soil solutions that is relevant for (i) more effectively designing stream water chemical analyses aimed at understanding the coupling of soil development processes and hydrology over variable time scales, and between deep and shallow weathering processes in mountain catchments; and (ii) for evaluating soil recovery processes after atmospherically induced perturbations that affected other catchments analogue to UDL."*

**RC2.13** The chapter on $^{18}O/^{16}O$ modelling in the Appendix seems not very well integrated in the study.

We added the following lines to the rewritten introduction to address this concern of the reviewer (page 3, lines 16-18): *"In addition, the contribution of groundwater vs. runoff infiltration is further evaluated by mean of a supplementary isotopic runoff model, which provides evidence for a likely contribution of groundwater enriched in selected chemical species due to sufficiently long water-saprolite interactions."*

Also the revised discussion text (page 12, lines 4-12) now reads: *"On this note, we followed the modelling approach implemented by Buzek et al. (1995, 2009) to provide further insight on the mean residence time of soil solutions—calculated across all sampling locations— which was estimated in approximately 8.3 months (Appendix A). In consequence, the runoff water at UDL is a mixture of direct precipitation with older soil solutions containing admixed with even older shallow groundwater. The supplementary isotopic modelling implemented here also shows that direct precipitation contributes between 20 and 40% of the discharge, with the rest being local soil pore and ground waters (Appendix A). The combination of all these three water types is called "mobile water", defined as the sum of all water pools and fluxes that respond to changing precipitation amounts. This mobile water transiently increases soil solution saturation and concomitantly with such an increase, the hydrologic connectivity of soil pore waters to the stream can cause a heterogeneous distribution of dissolved ions in soil solutions at the catchment-scale (Basu et al., 2010)."*

**RC2.14** The references are mostly relevant for the study and up-to-date. However, some citations aren't very specific to the referenced issues.

The reference have been revised as per reviewer indications (see below under Specific comments).

**RC2.15** The publication of the manuscript can only be recommend after a major revision considering the above mentioned concerns

We thank the reviewer for providing generous indications, guidance and sound revisions/ suggestions that significantly improved the original submission.

**R2. Specific comments**

**RC2.16** page 2, row 1 "Blazkova et al., 1996" -> "Blazkova, 1996"

Done

**RC2.17** page 2, row 3 "Blazkova et al., 1996" -> "Blazkova, 1996"

Done

**RC2.18** page 2, row 6 "Fen et al. 1998" doesn't appear in the references. Is "Fenn et al. 1998" meant? If so, this citation doesn't support the before stated sentence, because it mainly reflects the situation in North America.

Done.

**RC2.19** page 2, row 6 "Hruska et al. 2003" -> "Hruska and Kram, 2003"

Done.

**RC2.20** page 2, row 8 "Gradowski et al., 2008" -> "Gradowski and Thomas, 2008"

Done.

**RC2.21** page 2, row 9 "Matschullat et al., 1998" -> "Matschullat et al., 1992". This citation doesn't give relevant information on the bioavailability of phosphorus. Please replace accordingly.

Reference removed as per reviewer indication.

**RC2.22** page 2, row 10-11 This sentence isn't relevant for the introduction. It may be shifted to section 2.1.

Text was removed as part of the editions implemented to the Introduction.

**RC2.23** page 2, row 10-18 "For this aim, a nests of suction lysimeters was installed …" -> "For this aim, nests of suction lysimeters were installed …"

Done

**RC2.24** page 2, row 30 Would this kind of orthogneiss with the reported composition really be termed "alkaline"?

Word "alkaline" removed. The term apply to granite protolith over the base of the observed gneiss composition.

**RC2.25** page 3, row 1 "… m, UDL's …" -> "… m a.s.l., UDL's …"

Done

**RC2.26** page 3, row 2 At page 2, row 30, the parent material was named "alkaline gneiss". Is this the same as "porphyric granite"? Please clarify!

The first is the metamorphic product of the latter. We thank the reviewer for noticing the lack of clarity of the text. This has been now clarified (page 4, row 2-3): *"Low base status soils have developed at expense of the mineralogy of the porphyritic granite that served as protolith to the orthogneissic bedrock […]"*

**RC2.27** How can Podzols develop on "alkaline gneisses"?

The unclear statement was corrected as per note above. In igneous petrology, the composition of the igneous-metamorphic bedrock falls within alkaline granite in a compositional diagram.

**RC2.28** page 3, row 4 Are the given figures the runoff for one month? Please clarify!

We have clarified the text as per reviewer request (page 4, row 6): *" […] the monthly highest stream water runoff flow is usually recorded (~162 ± 29 mm)."*

**RC2.29** page 3, row 9 I wouldn't consider saplings trees up to 40 yrs.

True. The revised text now refers these to as *"trees < 40 yrs"* (page 4, row 8)

5 **RC2.30** page 3, row 12 Please delete "The."

Deleted.

**RC2.31** page 3, row 12-13 Please give the depth interval for the pH figures.

The revised text now reads (page 7, row 23): *"The soil at the 40-80 cm depth was characterized by pHH2O < 5 (Table 2)."*

10 **RC2.32** page 3, row 13 "… a pH increases in throughfall measurements …" -> "… a pH increase in throughfall …".

The revised text now reads (page 4, row 25-26): *"[…] while over the same period, median pH levels measured in water percolating through the canopy (i.e., throughfall) increased from 4.1 to 5.2 (Oulehle et al., 2017)."*

**RC2.33** page 3, row 13-14 Two times the same citation in one sentence isn't necessary.

15 The doubled citation referred by the reviewer is now removed from text.

**RC2.34** page 3, row 14-15 A comparison of pH(H2O) and pH(KCl) doesn't makes much sense.

Agreed. The offending text pointed out by the reviewer was removed.

**RC2.35** page 3, row 15 "Hruska, 2000" -> "Hruska, pers. comm." (unpublished data weren't published in 2000).

Reference removed as per note above.

20 **RC2.36** page 3, row 16-26 The content of this paragraph may be better placed in the results section. In addition figures presented should be given with standard deviation. Presented figures are a strange mixture of 13 or 14 catchments. Please revise consistently!

As suggested by the reviewer, a new section 2, titled *Study Site and Background Information*, now contains this information. There are 14 catchments in the Geomon network, this has been revised for consistency in page 4, line 25 32: "[...], which is far in excess of the atmospheric inputs observed in the remaining 13 GEOMON's monitored catchments across the Czech Republic".

**RC2.37** page 3, row 27 Section "2.2.1" follows "2.1". Please check for consistency.

Consistency of section and subsection numbering and general manuscript structure has been revised as per reviewer's suggestion.

30 **RC2.38** page 3, row 28 "… at a 50-cm depth below …" -> "… at 50 cm depth below …". Please change consistently throughout the manuscript.

Done.

**RC2.39** page 3, row 28 Were the lysimeters installed 50 cm below soil surface or below forest floor? Please clarify!

Done. The revised text now reads (page 5, row 6): *"were installed at 50 cm depth below soil surface"*

**RC2.40** page 4, row 6 the sampled depth intervals aren't horizons. Please specify!

Now these are referred to as soil levels (e.g., page 10, row 20; page 11, row 13).

**RC2.41** page 4, row 8 the given citation (FAO 2006) isn't about soil description. Please give correct citation!

FAO, 2006 removed from references.

**RC2.42** page 4, row 19 Please specify the relation of soil to water for pH measurement.

Done. The revised text now reads (page 6, row 5-6): *"A Radiometer TTT-85 pH meter with a combination electrode was used to measure $pH_{H2O}$ of soil (soil–water suspension ratio = 1 : 2.5)."*

**RC2.43** page 4, row 19-20 Please specify the meaning of soil moisture determination for the study, if any.

Any reference to soil moisture in text is now removed.

**RC2.44** page 4, row 22 "5 ° C" -> "5°C"

Done.

**RC2.45** page 4, row 23-27 The sequence of the determination of exchangeable cations seems in an incorrect order. Where NO3 and SO4 determined in the BaCl2 extracts?

We thank the reviewer for kindly pointing out this flaw in our original methods description. The text has been revised as follow (page 6, row 9-11): *"[…] The concentrations of $NO_3^-$ and $SO_4^{2-}$ from the soil extracts described above were determined by ion chromatography (HPLC Knauer 1000), with limit of quantification (D.L.) of 0.1 and 0.3 mg $L^{-1}$, […]"*

**RC2.46** page 4, row 28 Consider "For a phosphorus (P) release estimation, …" -> "For a estimation of phosphorus (P) availability, …"

Following the reviewer suggestion page 6, row 13 now reads: "For an estimation of phosphorus (P) release […]:

**RC2.47** page 4, row 28-29 Please specify the soil:solution relation and the concentration of the solution.

Done. The revised text (page 6, row 13-18) now reads: *"[…], ammonium oxalate extractions were performed by following the protocol described in Schoumans (2000). In short, a reagent solution consisting of $(COONH_4)_2 \cdot H_2O$ and $(COOH)_2 \cdot 2H_2O$ was used to dissolve 1 g of the <2 mm soil fraction. Extractable phosphorus ($P_{ox}$), iron ($Fe_{ox}$), and aluminium ($Al_{ox}$) were determined by shaking for 2 h in the dark duplicate samples of soils with 30 mL of 0.5 M reagent in 50 mL centrifuge tubes. […]."*

**RC2.48** page 5, row 1 "DPSox" in table 2.

Corrected as per Table 2.

**RC2.49** page 5, row 3 "Beauchemin et al., 1999" -> "Beauchemin and Simard, 1999"

Done

**RC2.50** page 5, row 4 "Borovec et al., 2018" -> "Borovec and Jan, 2018" page 5, row 11

Done

**RC2.51** Please introduce the meaning of D.L. at the first appearance! If detection limit is meant, consider to use "limit of quantification" instead.

Done (page 6, row 10): "*with limit of quantification (D.L.)*"

**RC2.52** page 5, row 14 Please explain the meaning of "non-parametric data".

Done (page 7, row 2): "The non-normally distributed (*i.e.,* non-parametric) data […]"

**RC2.53** page 5, row 14-22 The description of the factor analysis resembles - in my view as a nonstatistician – a parametric method. Please specify the non-parametric component of the statistical analysis.

Done. Please see note above.

**RC2.54** page 5, row 20 Please check the grammatical structure of the sentence.

The revised text now read (page 7, row 9-11): *"The variables can then be plotted in groups with correlation among them being determined by their position (e.g., proximity, distance, orthogonality,). The two-dimensional plane where the rotated, normalized data were plotted can be interpreted in terms of the main controls over the general variance of the dataset (see Vega et al., 1998 for details)."*

**RC2.55** page 5, row 24 The reporting of results for soil texture and pH in one subchapter appears strange to me. Why not report pH together with other soil chemical variables?

This flaw in our original masnucript' structure has been corrected (see new subsection in *4.2 Soil Chemical Characterization"*

**RC2.56** page 5, row 26-28 This sentence is more or less a repetition from section 2.2.1. Accordingly, it may be omitted.

The text was omitted as per reviewer's suggestion.

**RC2.57** page 6, row 1 Please give a definition of "pebbles" and "cobbles".

In the revised text (page 7, row 17) we now use "coarse soil particles (gravel and stones)"

**RC2.58** page 6, row 4-7 Consider to shift this part to section 3.3.

Moved to equivalent section 4.2.1 in the restructured manuscript.

**RC2.59** page 6, row 10 Is the mean given for all five sampling plots?

This revised text was clarified in this matter as follow (page 7, rows 28-29; 8, 1-2): *"In the eastern part of the catchment, the cation exchange capacity (CEC) of the mineral soil at 40-80 cm depth was up to 33 meq kg-1 on the slope and 58 meq kg-1 on the hilltop (Table 2). By contrast, in the western part, the CEC was 22 and 19 meq kg-1, which is lower than the mean CEC values measured at all of the plots at UDL, whilst CEC in the valley was 27 meq kg-1, which is within the mean CEC values measured at all of the plots at UDL: 32 ± 7 meq kg-1 (Table 2)."*

**RC2.60** page 6, row 16 Please check "twice larger" for correctness.

Done. Revised text (page 8 row 11-12) now reads: *"The BS at UDL is twofold higher than the BS determined at similar soil depths in the leucogranitic catchment LYS […]"*

**RC2.61** page 6, row 17 "Hruska et al., 2001" isn't in the references. Please check.

Revised. It is Hruska et al., 2000 (page 8, row 10)

5 **RC2.62** page 7, row 4-6 Water volumes haven't been mentioned before. This sentence may be omitted or the relevance of water volumes for the study should be emphasized.

This information was removed as is not further used in our results or discussions

**RC2.63** page 7, row 9 "… to be higher at …" -> "… to be highest at …"

Done.

10 **RC2.64** page 8, row 2 According to figure S2, explained variance is 24%

True. Corrected.

**RC2.65** page 8, row 3 According to figure S2, explained variance is 18%

True. Corrected.

**RC2.66** page 8, row 3 Is "Fig. S2" meant here?

15 Correct. Revised (now page 9, row 11)

**RC2.67** page 8, row 6-11 This part seems rather to be dedicated to the discussion. However, logical and grammatical consistency should be checked.

The text moved as per reviewer suggestion to discussion and also revised for grammatical correctness (page 12, rows 16-21): *"Finally, given the complexity of the possible interrelations among the environmental variables*
20 *considered here, there was an apparent generally poor correlation between solute concentrations measured in the soil in 2012-2013 and decadal runoff and atmospheric deposition data compiled in Table 1 (after Ouhlele et al., 2017). Such a result in turn points to a major control exerted by groundwater chemistry over soil solution chemistry, and also to soil organic and inorganic ligand properties also exerting a control over the residence time of each of the measured components."*

25 **RC2.68** page 8, row 7 Please explain the meaning of "apparent insignificant correlation".

See note above: *i.e.,* poor correlation

**RC2.70** page 8, row 23-24 "Manderscheid et al., 1995" -> "Manderscheid and Matzner, 1995"

Done.

**RC2.71** page 8, row 24 "Hruska et al., 2000" appears twice in the references. Which one is meant?

30 Corrected as per RC2.61

**RC2.72** page 8, row 24 "Armbruster et al., 2004" -> "Armbruster and Feger, 2004"

Done.

**RC2.73** page 9, row 9 "Meyer et al., 2001" is missing in references.

Done.

**RC2.74** page 9, row 12-14 Please check grammatical consistency of the sentence.

Done. Now page 11, row 14-16: *"The latter effect seems to be critical in the variability in soil solution chemistry at the hilltops, where the subsoil level contain significant amounts of coarse parental-rock material (Table 2)."*

**RC2.75** page 9, row 20 Is "Novak et al., 2005 meant here?

Right.

**RC2.76** page 10, row 4 What is meant with "areal control" here?

Text edited as follow (page 11, row 17-18): *"For Na+ and K+ ions in soil solutions, our spatially resolved time-series observations (Fig. 3) show that their concentrations defined patterns and trends largely derived from heterogeneity in soil granulometry (Table 2), with seasonality […]"*

**RC2.77** page 10, row 4-6 I can't follow this statement.

Corrected. Please see note above.

**RC2.78** page 10, row 8-9 What is the rationale behind the comparison of K and Na outputs?

The revised text (page 11) clarify the rationale and provide further information.

**RC2.79** page 10, row 14-15 To which other period is the spring season of 2013 compared here?

The request for clarification of the reviewer has been addressed as follow (page 11, row 29-30): *" […] a seemingly more rapid response of Na+ leaching to soil solutions could result from strong anion belowground episodic accumulation (Fig. 3; e.g., Spring 2013)*

**RC2.80** page 10, row 21 "Heuck et al., 2016" -> "Heuck and Spohn, 2016"

Done.

**RC2.81** page 10, row 23 "… in the 50 cm-depth …" -> "… in the 40-80 cm depth …"?

Done.

**RC2.82** page 12, row 5 "… these periods ware compared." -> "… these periods were compared."

Done.

**RC2.83** page 12, row 11 Figure A1a isn't visible in the manuscript. If "Figure 2A is meant, it should be corrected throughout Appendix A. page 12, row 26 Consider "… in soil pore spaces …" -> "… in soil pores …"

Done.

**RC2.84** page 12, row 28 What is meant with "direct precipitation"? If the "contribution of direct runoff (or "event water" as in eq. 2) to total runoff" is meant, a clear definition of "direct runoff" should be given.

The originally unclear text now reads "bulk precipitation" (page 15, line 1); "direct" removed

**RC2.85** page 15, row 21 "Fenn, E.M.;" -> "Fenn, M.E.;"

Reference removed.

**RC2.86** page 15, row 22 "Stottlemeye, R." -> "Stottlemyer, R."

Done.

**RC2.87** page 15, row 24-25 "FAO: Guideline for soil description; Rome, Italy, 2006" should be cited here.

Reference removed given that soil profile descriptions were done in previous works. See also reply to RC2.4.

**RC2.88** page 15, row 32-16/2 Please give correct title of the reference.

Done.

**RC2.89** page 16, row 3-7 "Hruska et al., 2000" appears twice. Please indicate the respective citations with "a" and "b". page 16, row 13 This line should be deleted.

Done.

**RC2.90** page 16, row 22 "Ma, L; Teng, F-Z.; Lin, L.:" -> "Ma, L; Teng, F.-Z.; Lin, L.; et al.:" page 18, row 2 "Soiling" -> " Solling"

Done.

**RC2.91** page 19, Figure 1 (b) Please consider to shift the sentence starting with "in the studied UDL …" to the text body of section 2.1.

Caption edited as suggested.

**RC2.92** page 23, Figure 2A The content of this figure relates to Appendix A and should be placed in the supplements accordingly?

Done.

[revised manuscript text omitted]

Due to pollution abatement policies, atmospheric input has decreased since peak acidification, yet UDL has been previously characterized by higher export of $SO_4^{2-}$, DOC, $Ca^{2+}$, $Mg^{2+}$, $K^+$, and $Na^+$ than their atmospheric input. In this regard, biogeochemical process within the soil seem to release more non-conservative ions than received from the atmosphere. Interestingly, export of total inorganic N from UDL *via* stream runoff continues to be significantly lower than its atmospheric input, but our results show that N leaching toward the subsoil levels is much higher than runoff. Toward the hilltops differences in porosity and greater fluid–rock-derived particle interactions, together with higher reactive surface area and solute flux , might as well exert a control over the measured soil solution chemical variability (Godsey et al., 2009). The latter effect seems to be critical in the variability in soil solution chemistry at the hilltops,  where the subsoil level contain significant amounts of coarse parental-rock material (Table 2).

For $Na^+$ and $K^+$ ions in soil solutions

~~Due to successful pollution abatement strategies, atmospheric input has decreased since peak acidification, and UDL has been characterized by higher export of $SO_4^{2-}$, DOC, $Ca^{2+}$, $Mg^{2+}$, $K^+$, and $Na^+$ than their atmospheric input. Conversely, export of total inorganic N from UDL *via* stream runoff continues to be significantly lower than its atmospheric input (Oulehle et al., 2017). As shown by Novak et al. (2004) using sulfur isotope ratios ($^{34}S/^{32}S$), cycling of the high amounts of deposited $SO_x$ at UDL occurred not only by adsorption/desorption of $SO_4^{2-}$ on soil particles, but, to a great extent, also by cycling through the soil organic matter, which may prevail for several decades (Novak et al., 2000; Armbruster et al., 2003; Mörth et al., 2005).~~

 our soil solution chemistries (Table 2; Figure 3).~~

5 ~~Within our time series, Ca²⁺ and Mg²⁺ concentrations in soil waters defined a general trend that likely reflect the balance between evapotranspiration and biological inputs, with a punctual, correlative shift recorded in concentrations measured during mid 2013 (Table 2). These seemed coeval to increased inputs in strong anions (Fig. 2). Increased leaching of these macronutrients could also be regulated by temporary changes in soil nitrate abundances (Oulehle et al., 2006; Wesselink et al., 1995; Akselsson et al., 2007, 2008). Whilst factor analysis did not reveal significant relationships between measured UDL~~

10

4.3 Retreat of Acidification

 spatially resolved, time-series observations (Fig. 3)  show that their  concentrations

15  defined patterns and trends largely derived from heterogeneity in soil granulometry (Table 2), with seasonality and pulses in atmospheric inputs also exerting some control over their concentrations  Fig. 2 and Fig. 3). For K⁺, and to a minor extent for Na⁺, soil solution concentrations recorded peaks that are more or less correlative to SO₄²⁻ and NO₃⁻ inputs (cf., Figs. 2 and 3), again pointing to lapses in which the atmospheric contributions of strong anions exerted a significant control over the weathering and leaching of plagioclase and K-

20 feldspar minerals in the underlying crystalline rock (e.g., Moore et al., 2012).  Oulehle et al. (2017) reported that K⁺ average annual  runoff was two to three times higher than that of Na⁺ (Table 1), with both cations exceeding runoff concentrations values measured in other monitored catchments.

When the  spatial variations in Na⁺ and K⁺ at the 50 cm- depth soil solutions  are also evaluated, a deep flow path within the eastern slope to the valley seems possibly augmented as a response of either atmospheric S inputs or

25 solubilization of the SO₄²⁻ stored in the weathering zone below the rooted soils due to soil water saturation . Because Na⁺ has low affinity toward organic and inorganic ligands in soil, and thus behaves relatively conservatively (McIntosh et al., 2017), a seemingly more rapid response of Na⁺ than K⁺ leaching to soil solutions could be interpreted as the result of the episodic accumulation of strong anions belowground  (cf. Fig. 2 and Fig. 3; e.g., Spring 2013). From this result, it can be argue that localized and punctual chemical analyses of

30 runoff waters in mountain catchments might not directly reflect nutrient partitioning trends along elevation gradients, but temporal variations of the strong anion content of the water table, which has implications for the design of studies centered in stream water analyses for understanding the coupling of soil development processes and hydrology over variable time scales and between deep and shallow weathering processes.

Such The observation behaviour of $Na^+$ vs. $K^+$ ions can in turnalso be interpreted as a decrease in water residence time from the slope to the valley-. On this note, we followed the modelingmodelling approach implemented by Buzek et al. (1995, 2009) to provide further insight on the mean residence time of soil solutions—calculated across all sampling locations— which was estimated in approximately 8.3 months (Appendix A)-, indicating that the volume of the entire mobile water at UDL is larger

5   than the volume of soil solution transported the 50 cm subsoil levels and below (see Appendix A). In consequence, the runoff water at UDL is a mixture of direct precipitation with older soil solutions containing admixed with even older shallow groundwater. The supplementary isotopic modelling implemented here also shows that direct precipitation contributes between 20 and 40% of the discharge, with the rest being local soil pore and ground waters (Appendix A). The combination of all these three water types is called "mobile water", defined as the sum of all water pools and fluxes that respond to changing

10  precipitation amounts. This effect is probably linked to mobile water transiently increased soil watersolution saturation and concomitantly with an increase in the hydrologic connectivity of soil pore waters to the stream, with acause a heterogeneous distribution of dissolved ions in soil solutions at the catchment-scale (Basu et al., 2010).

Whilst factor analysis did not reveal significant relationships between measured UDL parameters (Fig. S2), yet cross-plots in Figure 4 show a relatively strong pH–Mg/Al correlation. Both variables in each cross-plot reached the highest values on slope

15  east and the lowest values in the valley. Correlations seem to follow a spatial trend determined by the higher Al solubility at lower pH (Palmer et al., 2005). Finally, given the complexity of the possible interrelations among the environmental variables considered here, there was an apparent generally poor correlation between solute concentrations measured in the soil in 2012-2013 and decadal runoff and atmospheric deposition data compiled in Table 1 (after Ouhlele et al., 2017). Such a result in turn points to a major control exerted over the soil solution chemistry both by groundwater chemistry carrying legacy pollutants

20  and byover soil solution chemistry, and also to spatially variable soil organic and inorganic ligand-properties content, that also exerting a control overwhich likely determine the 
[revised manuscript text omitted]